# Particle Counting Methods Based on Microfluidic Devices

**DOI:** 10.3390/mi14091722

**Published:** 2023-09-01

**Authors:** Zenglin Dang, Yuning Jiang, Xin Su, Zhihao Wang, Yucheng Wang, Zhe Sun, Zheng Zhao, Chi Zhang, Yuming Hong, Zhijian Liu

**Affiliations:** 1College of Marine Engineering, Dalian Maritime University, Dalian 116026, China; d_ingenuity@dlmu.edu.cn (Z.D.); yuningjiang07@outlook.com (Y.J.); xin5231@dlmu.edu.cn (X.S.); yucheng_wang@dlmu.edu.cn (Y.W.); sunzhe1323162397@dlmu.edu.cn (Z.S.); zzheng@dlmu.edu.cn (Z.Z.); hym20000330@gmail.com (Y.H.); 2College of Marine Electrical Engineering, Dalian Maritime University, Dalian 116026, China; wangzhihao@dlmu.edu.cn; 3College of Transportation Engineering, Dalian Maritime University, Dalian 116026, China; forzhangchi@dlmu.edu.cn

**Keywords:** particle counting, microfluidics, sensitivity, throughput

## Abstract

Particle counting serves as a pivotal constituent in diverse analytical domains, encompassing a broad spectrum of entities, ranging from blood cells and bacteria to viruses, droplets, bubbles, wear debris, and magnetic beads. Recent epochs have witnessed remarkable progressions in microfluidic chip technology, culminating in the proliferation and maturation of microfluidic chip-based particle counting methodologies. This paper undertakes a taxonomical elucidation of microfluidic chip-based particle counters based on the physical parameters they detect. These particle counters are classified into three categories: optical-based counters, electrical-based particle counters, and other counters. Within each category, subcategories are established to consider structural differences. Each type of counter is described not only in terms of its working principle but also the methods employed to enhance sensitivity and throughput. Additionally, an analysis of future trends related to each counter type is provided.

## 1. Introduction

Particle counting is important for in situ or real-time analysis of samples in many fields, including biology [1,2], biomedicine [3,4], the environment [5], and engineering [6,7]. The particles here involve many types, such as blood cells, bacteria, viruses, droplets, bubbles, wear debris, magnetic beads, and so on. Microfluidic devices have shown enormous potential in particle counting in recent decades, with advantages such as low sample consumption [8], low time cost [9], portability [10,11], and so on. So, particle counting based on microfluidic devices has developed rapidly.

Integrating detecting techniques, like optical or electrical detecting techniques, with microfluidic devices has led to the development of various microfluidic counters based on different principles. A comprehensive review of these counters is essential to boost the microfluidic counter. Microfluidic detection devices have been reviewed extensively in the previous literature focusing on specific principles or applications [12,13,14,15,16,17]. Optical detection methods have been a popular choice, and Huo et al. [12] reviewed the advances in optical detection techniques applied in cell-based microfluidic systems. The study compared six optical detection methods and techniques, summarizing their respective trends, development perspectives, and advantages and disadvantages for label-free, real-time detection and sensing of living cells. Another popular technique is optical imaging, which combines microscopy with microfluidics. Zhou et al. [13] provided a comprehensive review of optical imaging systems, including bright-field microscopy, chemiluminescence imaging, spectrum-based microscopy imaging, and fluorescence-based microscopy imaging, and summarized the advantages and disadvantages of each technique. Ferrer-Font et al. [14] presented a spectral analyzer that enabled the concurrent analysis of up to 48 channels, thereby significantly enhancing the analytical capabilities of conventional flow cytometry systems. Magnetoimpedance-based biosensors have also gained popularity in recent years, and Wang et al. [15] reviewed the latest progress and achievements in this area. The study proposed constructive strategies for designing high-performance magnetoimpedance biosensors for magnetic-based counters, enabling the quantitative and ultrasensitive detection of magnetically labeled biomolecules. Electrochemical detection is another commonly used principle in microfluidic, which extends devices functionality of microfluidic chip counters. It can not only realize particle counting, but also provides chemical characteristic information of particles, realizes specific detection, real-time monitoring and particle characterization, and integrates with other functions of the microfluidic chip to expand the breadth and depth of particle analysis. Li et al. [16] provided a comprehensive review of recent advances and approaches to the design of microfluidic electrochemical systems for detection. The study discussed electrode and flow pathway design and highlighted some common challenges and solutions. Microfluidic-chip technology is also an emerging tool in the field of biomedical applications. Pattanayak et al. [17] discussed the design of various microfluidic chips and their biomedical applications. Mitchell et al. [18] conducted a comprehensive review of microfluidic devices for multiplexed biomarker detection. Their study primarily focuses on distinguishing and quantifying biomarkers using electrical and optical methods. Although there was also a systematic literature review of the microfluidic counters based on different principles in 2009 [19], a comprehensive review of microfluidic counters based on different principles is urgently needed, especially after 2009.

In this paper, microfluidic counters based on different principles are reviewed. The paper is structured as follows. As shown in Figure 1, firstly, optical-based counters, being the most widely employed method for particle counting, are thoroughly examined. Within the optical-based counters, four subclassifications are reviewed, namely fluorescence counters, optical absorption counters, scattering counters, and refractive index counters. Subsequently, the focus shifts to electrical counters, encompassing resistance-based counters, capacitance-based counters, inductance-based counters, and impedance-based counters. Finally, photoacoustic-based counters, magnetic-based counters, and thermal-based counters are comprehensively reviewed. For each counter type, a clear explanation of the underlying principles is provided, recent research findings are summarized, and strategies for enhancing sensitivity and throughput are discussed.

## 2. Optical-Based Counters

### 2.1. Fluorescence Counter

Fluorescence is the emission of light by a substance that has absorbed light, laser, or other electromagnetic radiation [20,21,22]. The fluorescence counter is a widely used detection technique due to its sensitivity and specificity. It works by exciting fluorescent compounds with a certain excitation wavelength and then detecting the emitted light, which has a longer wavelength, with little cross-interference from other matters. The principle of the fluorescence counter is shown in Figure 2a. The fluorescence counter typically consists of an excitation light source, objective lenses, dichroic mirrors, a microfluidic sample platform, a photodetector, and a data acquisition and processing unit [23,24]. As fluorescent particles pass through the optical window, they are excited by an appropriate wavelength of light, and the emitted light is detected by a photodiode and converted to electric signals [25]. These signals are then digitalized and processed to obtain particle counting results. 

The sensitivity of fluorescence counters is a crucial aspect of detection technology. In recent times, there have been many studies aimed at improving the sensitivity of these counters. There are mainly two methods to achieve this. The first approach involves enhancing the intensity of the fluorescence emitted from particles using micro-optical lenses, a special molecular probe, and better fluorescent markers. Lim et al. [26] improved the fluorescence signal by eight times by designing a droplet-based microfluidic device that integrated microlenses and mirror surfaces to enhance the signals, as shown in Figure 2b. Cao et al. [27] introduced an optofluidic platform to enhance fluorescence collection from a microfluidic channel. The detection module includes a monolithic parabolic mirror positioned above the channel, increasing the number of emitted photons directed towards the detector. This approach demonstrates a fluorescence signal enhancement of up to 113-fold. Except using micro-optical lenses to enhance fluorescence intensity, Gallina et al. [28] described a new concept that used a molecular probe sensitive to ionizing radiation byproducts to convert random radioactive decay into a long-lasting fluorescent signal, resulting in an enhanced fluorescent signal. The use of better fluorescent markers, such as quantum dots (QDs), is another way to enhance the fluorescence signal. QDs have many advantages over organic fluorophores, including a broad excitation range, narrow emission bandwidth, high quantum yield, and exceptional photochemical stability. Zhang et al. [29] proposed a single-QD-based nanosensor with near-zero background noise that significantly improved the sensitivity of microRNA assays by up to two orders of magnitude. The second approach involves reducing the noise of the counter. More importantly, QDs of different sizes are excited by the same wavelength of light to obtain multiple color labels, making them ideal probes for multiplex analysis. Yin et al. [30] developed an immunosensor using QD-reverse detection strategy and immunomagnetic beads for simultaneous detection of *Escherichia coli* O157: H7 and Salmonella. This approach minimizes interference, boosts fluorescent signals, and streamlines the process. Compared to traditional QDs immunosensors, detection of Escherichia coli O157: H7 improved by 50 times, with a 30 cfu/mL limit of detection, and analysis completed within an hour.

Recent research has shown a growing interest in the development of high-throughput microfluidic counters [24]. To this end, Li et al. [31] presented an optical flow cytometer (OFCM) for single-cell phenotype counting. It uses a four-color fluorescence detection system that can simultaneously detect the four-color fluorescence of individual circulating tumor cells (CTCs) flowing through the detection channel of the chip and obtain information on the expression of multiple phenotypic markers. This system enables rapid and accurate classification and counting of CTCs with a throughput of 1.2 mL of whole blood per hour, as shown in Figure 2c. Differently, Cao et al. [27] proposed an optofluidic platform for enhanced collection of fluorescence from a microfluidic channel, which effectively improved the throughput by detecting pL-volume droplets at a rate of up to 40,000 droplets per second. Additionally, by combining parallelized droplet production strategy and time-domain encoded optofluidics, Yelleswarapu et al. [32] created a microdroplet megascale detector capable of achieving ultra-high throughput droplet detection of up to 103,000 droplets per second using 120 parallel channels and a conventional cell phone camera. Their innovative approach involves modulating the excitation light using a pseudorandom sequence to resolve individual droplets that would otherwise overlap due to digital camera frame rate limitations. Additionally, Kim et al. [33] developed a counter composed of 16 parallel microfluidic channels directly bonded to a filter-coated two-dimensional Complementary Metal-Oxide-Semiconductor (CMOS) sensor array. The device enables the counting of fluorescent drops at a throughput of 254,000 drops per second, constrained by the acquisition speed of the CMOS sensor.

**Figure 2 micromachines-14-01722-f002:**
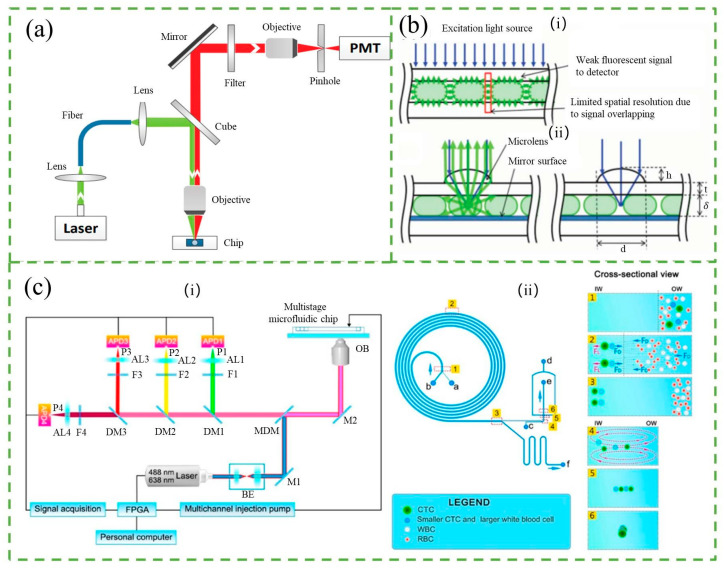
(**a**) Schematic of fluorescence detection system. Reproduced with permission from Ref. [34]. Copyright © 2023 AIP Publishing. (**b**) (**i**) Limitations of conventional droplet-based microfluidic device. (**ii**) Schematic diagram of micro-optics integrated droplet-based microfluidic device. Reproduced with permission from Ref. [26]. CC BY license. (**c**) (**i**) Schematic diagram of optical path and overall composition of the OFCM. (**ii**) Structure diagram of multistage microfluidic chip. Reproduced with permission from Ref. [31]. Copyright © 2023 American Chemical Society. The instrument comprises a multistage microfluidic chip, a four-color fluorescence detection module, a signal acquisition and data processing module, and a system control module. “a”, “b”, “c”, and “d” denote the sample entrance, sample sheath entrance, vertical sheath entrance, and horizontal sheath entrance, respectively. “e” and “f” represent the CTC exit and waste liquid exit. Inserts 1–3 and 4–6 depict the schematic diagrams of CTC separation and 3D focusing, respectively.

The accuracy of fluorescence counting is a crucial aspect to be considered in many applications. To address this issue, Liu et al. [35] developed an immunosorbent assay for the digital identification of target exosomes using droplet microfluidics. This approach allows for the absolute counting of cancer-specific exosomes, resulting in unprecedented accuracy. The droplet-based single-exosome-counting enzyme-linked immunoassay (droplet digital ExoELISA) method immobilizes exosomes on magnetic microbeads via sandwich ELISA complexes and labels them with enzymatic reporters to produce a fluorescent signal. The developed method reduces the limit of detection (LOD) to 10 enzyme-labeled exosome complexes per microliter (10^−17^ M), allowing for the discovery of cancer exosome biomarkers for clinical diagnosis and potentially enabling early cancer detection [36]. Another study by Xia et al. [37] created a cyclic olefin copolymer (COC) microchip with a simple cross-channel layout that, once coupled to a laser-induced fluorescence detector, resulted in a microfluidic cytometer with a counting accuracy of 97.6%. This improved accuracy allowed for more precise counting of lower concentrations of large microspheres. The developed microfluidic cytometer is simple to design and manufacture, easy to operate and optimize, and shows potential for clinical applications.

Fluorescence counters are widely used in biomedical applications due to their sensitivity and ease of integration with microfluidic chips. These applications include cancer diagnosis and detection of live CTCs, and so on [31,36,38,39]. However, a significant drawback is that particles must exhibit fluorescent characteristics, or nonfluorescent particles must be labeled with proper fluorophores.

### 2.2. Optical Absorbed Counter 

In addition to fluorescence counters, there are non-fluorescence counters such as absorbed counters that detect light intensity after passing through particles. While fluorescence detection offers high sensitivity for small volume droplet analysis, it has limited use due to the requirement for intrinsic fluorescence or extrinsic fluorophore labeling. In contrast, absorption spectroscopy allows for both label-free and quantitative analysis and is universal for almost all organic compounds with conjugated functional groups [20]. The principle of the absorbed counter is depicted in Figure 3a, where a laser or other light source passes through the particle in the microchannel after being adjusted by lenses, and the decrease in light intensity is detected as it passes through the particle.

The sensitivity of the absorbed counter is a crucial factor in microfluidic detection methods, just like the fluorescence counter. Mao et al. [42] developed a droplet-based optofluidic system with microfluidic droplet generation and optical fiber-based absorption spectroscopy detection. They embedded two optical fibers orthogonal to the longitudinal axis of the microfluidic channel to detect Rhodamine 6G at concentrations down to 0.1 mM in flowing droplets. In fact, the absorbance value is proportional to the length of the light path, which is proportional to the channel depth. However, the short light path length of conventional shallow channels is considered a disadvantage for absorption spectroscopy [43,44]. To address this issue, Onoshima et al. [45] introduced a deeper microfluidic counter by using dry film resist SU-8 to create a thick mold for soft lithography of a polydimethylsiloxane (PDMS) microfluidic chip with deep channels. This technique achieved a detection limit (15.9 μM) comparable to microfabricated absorbance detection cells in glass. Additionally, Yang et al. [40] integrated a liquid-core PDMS waveguide into a droplet chip to accomplish detection limits of 400 nM and sensitive absorbance measurements of picoliter (pL) droplets at a throughput of 1 kHz, as shown in Figure 3a. By employing a Z-shaped detection path, the accessible optical path length was significantly increased, improving absorption measurement sensitivity. Probst et al. [41] have proposed a novel photofluidic platform for sensitive acquisition of broadband absorption spectra from pL volume droplets, as shown in Figure 3b. The use of confocal illumination effectively limits the detection region to a size close to that of the contained droplet. Anyway, microsecond spectral acquisition combined with a simple post-processing scheme effectively removes the oil contribution and light source instabilities that contribute to background instability that contribute to improving sensitivity.

Except the sensitivity, miniaturization, accuracy, and high throughput are also involved in research on the absorbed counter. To achieve miniaturization and simplify the detection system, Lamprecht et al. [46] proposed combining monolithically integrated ring-like sensor waveguides with ring-shaped thin-film organic photodiodes on a single substrate. Another study by Wu et al. [47] introduced a multifunctional cell counting microdevice utilizing a center-pass optofluidic microlens array to guide 6∼8 μm width cells to pass through the edges of adjacent microlenses. This approach enabled successful parallel cell counting without coupling issues, achieved by monitoring optical intensity variations at each spot. Moreover, droplet-based absorbance counters have been developed for high-throughput applications. Gielen et al. [48] designed a microfluidic sorter based on an absorbance counter, where a decrease in transmittance relative to the dye concentration was recorded during droplet passage through the interrogation volume, serving as the basis for sorting decisions. Their device was capable of sorting up to 300 droplets per second. Banoth et al. [49] developed a technique that simultaneously detects changes in cell morphology and chemical composition. This technique allows for higher throughput of approximately 1000 cells per second and can detect low levels of parasitemia in blood samples within a comparable time frame to rapid diagnostic tests.

The optical absorbed counter represents one of the most common detection methods employed in microfluidics, offering the advantage of being label-free and straightforward to implement. A broad range of wavelengths, from blue to UV/Vis and infrared spectra, can be exploited for absorbance measurements, enabling a variety of applications. However, the optical absorbed detector’s performance may be limited in cases where the solution’s transparency is low. Despite this, the optical absorbed counter has demonstrated the ability to measure the concentration of analytes in droplets or bioparticles, as evidenced by various studies [40,45,50,51]. For instance, malaria has been detected by measuring the single-cell optical absorbance levels of different cell types in the blood, as reported in previous research [49,52,53]. 

### 2.3. Scattering Counter 

When the source light interacts with the particle, some of the light is scattered, and the resulting scattered light can be detected by a photo-detector placed at a specific angle from the light path. The detection of the scattered light is commonly achieved through the use of forward scatter (FSC) or side scatter (SSC) measurements, as illustrated in Figure 4a. For example, Zhao et al. [54] designed a microfluidic cytometer featuring an integrated on-chip optical system for red blood cell and platelet counting. The device was fabricated using a single-mask process and standard soft lithography techniques, as depicted in Figure 4b. A simultaneous collection of forward scatter (FSC) and extinction signals was performed for each cell. The compact structure of the device improves sensitivity and overcomes the problems of bulky, expensive, and rigid fixation. Detectors are typically placed at an angle of approximately 20–40° relative to the particle beam path. The magnitude of the detected optical signal corresponds to the size of the particle, while the number of pulses in the signal is proportional to the particle number [19].

Many studies have focused on enhancing the sensitivity of detecting light scattering in microfluidic chips. To achieve this, it is crucial to ensure that particles flow across the beam of light to generate a strong enough scattering signal for detection. As a result, particles require 2D or 3D focusing before entering the detection window. Guo et al. [56] developed a scattering counter that employed a 2D hydrodynamic sample focusing to count *E. coli* DH5α-cell suspensions in PBS solution, achieving a detection efficiency of 92% without tagging the cells. Dannhauser et al. [57,58] proposed a small angle light scattering apparatus that used 3D focusing of particles to collect FSC light emitted by the particles, enabling detection of particles as small as 1 µm in radius and multiplex applications. Three dimensional focusing generally allows higher particle densities within the detection area, resulting in increased signal strength. Nonetheless, achieving 3D focusing involves more complex optical setups and fluidic manipulations, which sometimes limits their usefulness. Comparatively, 2D focusing, while marginally less sensitive, offers feasibility in high-throughput setups. In addition, the optimization of chip materials and system structures is also conducive to improving sensitivity. Shivhare et al. [59] designed a PDMS chip with predefined notches where FSC signals were analyzed to determine scattering signal intensities and residence times, which were correlated with droplet number and droplet size. However, the presence of random scattering signals from rough surfaces in the chip material like PDMS can cause high signal noise, which can mask the actual signal intensity and lower sensitivity. To overcome this issue, it is effective to use chips with smooth surfaces, reduce refractive index variation, and keep the fiber close to the detection channel. Furthermore, reducing the frequency of the droplet can provide the sensor with a longer acquisition time, which improves measurement quality while reducing throughput [60]. In contrast, Kwon et al. [55] increased the volume of particles to count smaller particles in a compact and cost-effective detection system based on condensation nucleation light scattering technology, as shown in Figure 4c. This system can optically count individual nanoparticles (NPs) as small as 9.3 nm over an extremely wide concentration range (0.021–10^5^ N cm^−3^) with high accuracy, using water as the condensing liquid, thereby addressing the self-contamination problems associated with most portable NP detection systems.

Among scattering counters, there is a special counter named the Raman counter. It is worth noting that this type of scattering always has a different frequency than that of the incident photons. The surface-enhanced Raman scattering (SERS) is an ultra-sensitive spectroscopic detection technique that employs a metal surface to enhance the Raman signal of the analyte, leading to a remarkable amplification in the spectral signal of trace molecules. By using SERS-active nanoparticles, the intensity of Raman signals can be increased at least several times. For instance, Szymborski et al. [61] employed dielectrophoresis (DEP) to separate circulating tumor cells (CTCs) and deposit them on a surface-enhanced Raman spectroscopy (SERS) platform. This technique enhanced the spectral resolutions and intensities of specific marker bands by up to 13-fold. Freitag et al. [62] demonstrated the detection of tumor cells using immuno-SERS markers in a microfluidic chip with continuous flow, enabling rapid and reproducible SERS-assisted cell detection. The amplified Raman signals facilitated the identification of analytes at lower concentrations, reduced collection time, and increased throughput. Moreover, droplet microfluidics can leverage other nanoparticles for Raman signal enhancement [63,64]. Nonetheless, challenges in Raman spectroscopy include contamination peaks in chip materials and high laser power requirements [20,60]. Integration of optical fibers can mitigate spectral background interference from chip materials like PDMS, as the fibers can be embedded near the analyte, while longer acquisition times can enhance detection sensitivity.

The scattering counter offers the benefits of being label-free and easy to integrate with microfluidic chips. It is capable of distinguishing between different types of particles, making it useful for analysis. However, minimizing the counter size remains a challenge due to the need for certain optical components.

### 2.4. Refractive Index Counter

Refraction is the phenomenon of the change in direction of wave propagation due to a change in its transmission medium. This effect is explained by the principles of energy and momentum conservation. When a wave passes from one medium to another at any angle other than 0° from the normal, the phase velocity of the wave is altered, while its frequency remains constant. This concept is frequently observed in various fields, including optics, geophysics, and acoustics. Based on this fundamental principle, the refractive index (RI) counter has been developed as a useful tool for counting cells, bubbles, or droplets. 

To improve the sensitivity of the RI counter, various novel designs have been proposed. Erin et al. [65] introduced a planar, chip-based, dual-beam refractometer, which utilized a central organic light-emitting diode (OLED) light source and two organic photovoltaic (OPV) detectors on an internal reflection element (IRE) substrate. This configuration provided two sensing regions, a “sample” and a “reference” channel, and allowed for enhanced sensitivity to RI changes. In comparison to single-beam operation, the dual-beam configuration could detect changes in the refractive index (ΔRI) with an increase in sensitivity of at least one order of magnitude. Moreover, Xing et al. [66] developed an ultrasensitive graphene-based optical RI sensor by controlling the thickness of high-temperature reduced graphene oxide (h-rGO), as shown in Figure 5a. At the single-cell level, it enables label-free, live-cell, and highly accurate detection of a small number of cancer cells among normal cells. By optimizing the thickness of h-rGO, they achieved a higher limit of sensitivity and resolution for RI sensing, up to 4.3 × 10^7^ mV/RIU and 1.7 × 10^−8^, respectively. In another study, Yan et al. [67] designed and fabricated an RI sensor based on an optical flow arch waveguide structure, as shown in Figure 5b. By monitoring the power loss of the light passing through the waveguide, which is sandwiched between the air-cladding and the liquid-cladding under test, it demonstrated a high sensitivity of −19.2 mW/RIU and a low detection limit of 5.21 × 10^−8^ RIU.

Thermal lens microscopy (TLM) is a highly sensitive detection technique based on the principle of RI counters. Initially used by Sawada’s group [69], TLM was used to count nanoparticles, detecting 80 nm polystyrene particles and 10 nm Ag particles in water. TLM involves coaxially aligning an excitation beam and a probe beam, which passes through the objective lens of the microscope and is focused on the solution within the microchannel. In the absence of nanoparticles in the detection volume, the probe beam traverses the volume without deviation. However, when nanoparticles are present, they absorb laser photons and release energy into the liquid, increasing its temperature. Since liquids generally exhibit a negative temperature coefficient of refractive index (dn/dT), the solution’s refractive index in the center of the excitation beam becomes lower than that of the surrounding solution. Consequently, a spatial temperature gradient acts as a concave lens, resulting in the thermal lens effect. This effect causes the deflection of the probe beam, leading to a deviation in light intensity after passing through the pinhole. The sensitivity of TLM to nanoparticles is higher than that of an absorption measurement because the former measures deviations in light intensity while the latter measures decreases in excitation intensity. 

To improve the sensitivity of TLM, researchers have developed two specific measures to ensure that particles pass through the focusing spot of the probe beam, which is typically about 1 µm in size. One approach involves fabricating a 1 µm scale microchannel using electron-beam lithography and dry etching, as demonstrated by Seta et al. [70]. They were able to detect an individual, label-free nanoparticle with a diameter of 130 nm using this method. Another approach involves using flow focusing. Yamaoka et al. [68] fabricated a microfluidic device that sandwiched the sample flow between lateral z-axis sheath flows and horizontal xy-plane sheath flows, resulting in a three-dimensional flow focus, as shown in Figure 5c. This method allowed for the detection of 500 nm sized polystyrene particles. In addition, Mahdieh et al. [71] investigated the effect of diode laser power on beam quality when focusing. They induced a thermal lens effect by focusing the diode laser beam into ethanol and found that the beam quality factor M^2^ of He-Ne increased non-linearly as the diode laser power was increased. While high laser power can decrease the beam quality, it is necessary to increase the laser power as much as possible without compromising the quality in order to induce the thermal lens effect. TLM is a highly sensitive technique that can be used to detect the concentration of nanoparticles without the need for labeling [72]. However, it is important to ensure that the sample solution is suitable and that environmental effects are minimized.

The RI counter is a powerful tool for label-free detection of single cells with high accuracy. In addition to cell counting, it can also monitor the state of cells, making it useful in a variety of applications. However, it is important to note that RI counters are mainly used for analytic concentration detection, and may not be suitable for counting opaque particles such as wear debris or magnetic beads [73,74]. It is also important to control environmental factors such as temperature, pressure, and flow rate during counting, as these can affect the accuracy of the results [75]. Overall, the RI counter is a valuable tool for cell analysis, but careful consideration of its limitations and experimental conditions is necessary for accurate and reliable results.

## 3. Electrical-Based Counters

The Coulter Counter is a simple yet effective type of electrical counter that was first invented by Wallace H. Coulter during World War II and patented [76]. It typically consists of two chambers, an inlet and outlet reservoir, which are separated by a single microchannel. As particles pass through the microchannel, they can cause changes in the resistance, capacitance, or inductance of the counter integrated with the microchannel. These changes can be detected as voltage or current pulses, providing information about the particles’ number, size, shape, mobility, and surface charge. The Coulter Counter is a versatile tool for particle characterization, but it has limitations such as the inability to differentiate between particle types and difficulty in analyzing complex samples.

### 3.1. Resistance-Based Counter

The resistive-based counter, also known as the resistive pulse sensor (RPS), operates on a principle similar to the Coulter Counter. When a fluid carries a particle through a micro- or nanoscale aperture, the aperture’s resistance significantly increases due to the blockage caused by the particle [77]. This blockage results in a significant change in the electrical current across the aperture, producing a pulse. Particle counting is accomplished by detecting these current pulses using an external circuit. 

Various approaches have been explored to improve sensitivity, including the use of focused flow and new structural systems and techniques [78,79,80,81,82,83,84,85]. A popular approach to focusing stream has been the use of non-conducting solutions such as pure water. The effective aperture size decreases with the presence of a focusing stream, allowing for the relative current pulse caused by the particle to be amplified. Liu et al. [78] introduced a novel electrokinetic focusing method that used a high-resistivity focusing solution flowing from upstream to downstream channels to improve detection sensitivity, as shown in Figure 6a. This method narrowed the sensing gate, greatly improving sensitivity to detect particles as small as 1 μm with a sensing gate of 30 × 40 × 10 μm (width × length × height). Other researchers have also adopted this method, which can be driven by both electrodynamics and pressure [86,87,88].

There are also many new structural systems and techniques adopted to improve sensitivity, such as differential RPS, reducing sensing orifices, and multiple pores in series. Differential RPS, for example, employs two detecting electrodes instead of only one, with one detecting the upstream of the aperture and another detecting the downstream, to reduce noise. Song et al. [81] reported on a lab-on-a-chip device that used differential RPS to count the number of bacteria flowing through a microchannel, achieving a signal-to-noise ratio (SNR) of 5–17 for the detected RPS signal amplitude. Peng and Li [82] were able to detect nanoparticles as small as 23 nm using differential RPS on PDMS nanofluidic chips. Song et al. [79] proposed a new side-orifice-based RPS (SO-RPS) for the detection of nanoparticles and microorganisms, as shown in Figure 6b. The detection sensitivity of the SO-RPS was improved by reducing the channel height in the detection section, leading to an average SNR of approximately three for 100 nm polystyrene particles. They also counted particles by local DC dielectric electrophoretic forces, successfully separating and counting two and three different sizes of polystyrene particles with 1 mm resolution. Zhang et al. [80] proposed a novel device with five identical sensing regions connected in series and multiple cross-correlation analysis that enhances the sizing SNR by a factor of n^1/2^, where n is the pore numbers in series, as shown in Figure 6c. Additionally, a novel signal averaging algorithm has also been developed by Ashley et al. [83,84] that reduces noise in microfluidic impedance cell counting data, improves enumeration accuracy, and reduces detection limits. These innovative technologies have significantly improved the sensitivity and accuracy of particle detection and counting in microfluidic systems. In addition to increasing the sensitivity, there is research that makes the sensitivity easily tunable. Platt et al. [85] designed a tunable resistive pulse sensing (T-RPS) device with a reusable lid and base that allows for two ways of adjusting the size of the pore to the size of the analyte in real time. T-RPS is unique in that the membranes are elastomers, allowing for in situ optimization of the SNR of the resistive pulse signal. Particle sizes measured using T-RPS range from microns to ~50 nm, covering a length scale of approximately two orders of magnitude and bridging the gap between single molecules and cells.

Increasing throughput is crucial for RPS, and multichannel detection methods are commonly used to achieve this. Jagtiani et al. [89] developed an on-chip multiplexed multichannel resistive pulse sensor for high-throughput counting of microscale particles. The device employed multiple parallel microfluidic channels for sample analysis, using frequency division multiplexing for detection. Each microchannel was modulated with a unique known frequency, and a combined measurement was obtained across a single pair of electrodes. The measured signal was then demodulated to determine the signal from each individual channel. Testing with 30 μm polystyrene particles demonstrated that the multiplexed device achieved a 300% higher throughput compared to a single-channel device, as it allowed simultaneous detection of particles through its four parallel channels. Song et al. [90] achieved high-throughput particle counting in a microfluidic chip using a differential RPS with multiple detecting channels. They developed a sensitive differential microfluidic sensor with multiple detecting channels and one common reference channel and achieved an average throughput of 7140/min at a flow rate of 10 µL/min. In contrast to multichannel detection methods, SO-RPS is a recent technology that uses sensing orifices located on the sidewalls of microchannels for particle detection and counting. The SO-RPS proposed by Song et al. [79] also easily achieved a high flow rate or high particle throughput under a low-pressure difference since the sample solution does not need to pass through the orifice so that could avoid channel clogging. 

Microfluidic RPS has several advantages, such as label-free particle detection, simplicity, and requirement of only a simple circuit and a micron- or nanometer-sized channel [88]. Hole-based resistive pulse sensors, in particular, have been widely used to detect, measure, and analyze particles ranging in length from nanometers to micrometers due to their simplicity and robustness. However, RPS cannot work with non-conductive solutions (such as oil) since the solution in the sample must be conductive to ensure that it can create a reliable circuit between the sensor electrodes. While smaller sensing aperture diameters can lead to a higher detection sensitivity, they can also lead to a lower throughput and higher differential pressures as well as channel clogging when the aperture diameter is similar to the particle diameter. So, there are some studies on how to avoid channel clogging [79,89,90]. In addition, various studies have been conducted on cost-effective RPS [17,91,92], miniaturized RPS [93], and others. 

### 3.2. Capacitance-Based Counter

As mentioned above, the RPS method is limited to conductive sample solutions. When dealing with non-conductive samples, the capacitance-based counter is a better option. Similar to RPS, the capacitance-based counter also detects changes in capacitance between two electrodes in the microchannel. The principle of the capacitance-based counter is shown in Figure 7a. 

Researchers are currently working on improving the sensitivity of the capacitance-based counter. Do et al. [94] developed a differential capacitive coupled contactless conductivity detection sensor with a thin PDMS protective layer. The capacitive sensor consists of four adjacent electrodes arranged to form differential coplanar capacitor structures, which provides high sensitivity and robust operation. The differential capacitance changes when a microparticle (such as living cells) passes through the microfluidic channel. The simulation inspection showed that the sensor could detect an object with a diameter as small as 10 µm in a 50 × 100 µm cross-section channel, with a capacitance change of up to 0.05 fF. Shi et al. [95] described an integrated sensor consisting of an inductance-resistance sensing unit and a capacitance sensing unit for detecting multi-contaminants in hydraulic oil, as shown in Figure 7b. Capacitance parameter is used to measure air bubbles and moisture, and the moisture measurement is not interfered by solid impurities. Bubbles of 95 μm were successfully detected in the capacitance experiments, and high sensitivity counting was achieved. Similar to the method of condensation nucleation described in Section 2.3 [55], Jeon et al. [96] proposed a method based on condensation nucleation that counts particles according to their dielectric constant, as shown in Figure 7c. The experiments showed that the system can grow particles larger than 50 nm into micron-sized droplets with linear electrostatic properties of up to 10,300 N cm^−3^, effectively reducing the size and cost of the system.

To further explore the application of the capacitance-based counter, Murali et al. [97,98] demonstrated the feasibility of using the capacitance-based counter to detect and count micrometal particles in nonconductive lubricant oil. Additionally, Barbosa et al. [99] developed a capacitive bubble counter to measure two-phase flows, and Song et al. [100] designed a three-dimensional capacitance sensor to detect living microalgae. It is clear that the capacitance-based counter has a wide range of applications, from detecting metallic debris in hydraulic fluid to counting microalgae in a microfluidic chip.

While the capacitance-based counter offers the advantage of particle counting in non-conductive solutions, it is not without limitations. One major challenge is the complexity involved in electrode fabrication, which can lead to difficult and time-consuming manufacturing processes. Furthermore, the sensitivity of capacitance-based counters is not always high, which may limit their accuracy in detecting and counting small particles. To address these limitations, researchers are actively exploring ways to improve sensitivity and simplify electrode fabrication for capacitance-based counters.

**Figure 7 micromachines-14-01722-f007:**
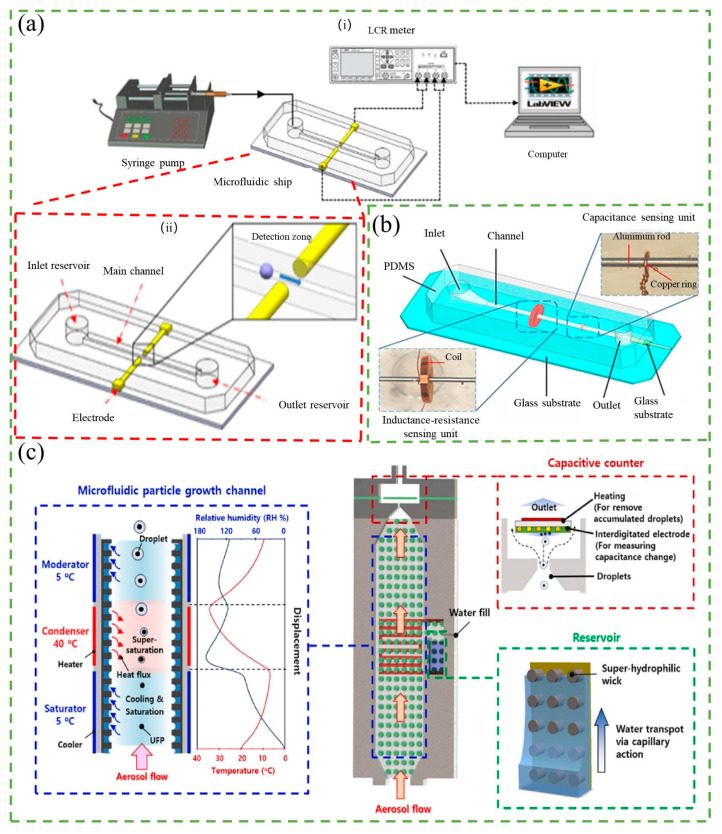
(**a**) Schematic diagram of the chip structure (**i**) and the capacitive detection system (**ii**). Reproduced with permission from Ref. [100]. Copyright © 2023 Elsevier B.V. (**b**) Schematic diagram of the inductance-resistance-capacitance (IRC) sensor. Reproduced with permission from Ref. [95]. Copyright © 2023 Elsevier Ltd. (**c**) Schematic diagram of the condensation nucleation-based system, including a microfluidic particle growth channel, capacitive particle counter, and reservoir. Reproduced with permission from Ref. [96]. Copyright © 2023 Elsevier Ltd.

### 3.3. Inductance-Based Counter 

When metallic particles pass through the channel, they change the inductance of the coil that is just outside the channel. The variation of the inductance could be detected, and that is the principle of the inductance-based counter. Du et al. [101] firstly presented a microfluidic device based on the inductive Coulter counting. They tested the device with iron and copper particles ranging from 50 to 125 µm and demonstrated its ability to detect and distinguish ferrous and non-ferrous metal particles in lubrication oil, as shown in Figure 8a. 

To improve the sensitivity of microfluidic inductance-based counters, researchers have explored various methods including reducing the inner diameter of the coil and using novel materials, to enhance the magnetic field strength, such as silicon steel, magnetic powder, and permalloy. Shi et al. [105] used a strip of silicon steel to enhance the magnetic field strength in the detection area of an inductive sensor’s planar coil. They found that the integrated sensor with 300 μm channels could effectively detect and differentiate air bubbles larger than 80 μm, iron particles of 30–300 μm, and copper particles of 45–300 μm. They also designed a new integrated sensor using a ring-shaped silicon steel sheet wrapped around the outer layer of the planar coil [102], which was able to detect iron particles of 55 μm and copper particles of 115 μm by combining the inductive signal with the resistive signal from a 20-turn flat coil, as shown in Figure 8b. These approaches offer promising strategies to enhance the sensitivity of inductance-based counters for detecting non-ferromagnetic metal particles. Bai et al. [106] used magnetic nanoparticle materials to make inductive oil detection sensors, resulting in a higher signal-to-noise ratio (SNR) for detecting ferromagnetic and non-ferromagnetic particles. The detection SNR for 20–70 μm ferromagnetic particles was improved by 20–25% and 80–130 μm non-ferromagnetic particles was improved by 16–20%. Liu et al. [107] proposed a method of placing magnetic powder around the inductor coil to improve the sensitivity of a miniature inductive sensor to detect abrasive grains. The method could extend the detection limit of the micro inductive sensors used in the experiments to 11 µm abrasive particles. Wang et al. [103] designed a highly sensitive inductive sensor for detecting debris in lubricating oil using the mutual inductance of coils and the strong magnetic permeability of a permalloy sensor, as shown in Figure 8c. The permalloy has high permeability, low coercivity, high saturation magnetization strength, a sensitive response to weak magnetic signals, and good magnetic shielding. The sensor could detect 10–15 µm of iron particles and 65–70 µm of copper particles in the oil. Similar to the differential resistive pulse sensor (RPS), Yang et al. [108] proposed an inductive sensor based on differential detection that employs two induction coils embedded in a single excitation coil. By reverse connecting the two induction coils with identical parameters, the differential signal is obtained, thereby suppressing common mode interference and eliminating ambient noise effects to produce an ultra-low noise sensor. The experimental results revealed that the sensor can detect 20 μm iron particles and 130 μm copper particles in a 2 mm flow channel, with a detection error of less than 22%.

In order to improve the throughput, a variety of multichannel sensor systems based on different principles have been proposed, such as frequency division multiplexing, phase division multiplexing, and time division multiplexing. Du et al. [109] firstly presented an inductive pulse sensor based on resonant frequency division multiplexing, in which each inductor coil operates at a different excitation frequency. The sensor comprises four inductive coils connected in series, each in parallel with a capacitor to create a parallel resonant circuit with a specific resonant frequency. This four-channel sensor can detect multiple coil signals using a single excitation signal and one signal acquisition channel, resulting in a 300% increase in throughput. Differently, Wu et al. [110] proposed a multichannel wear debris sensor based on phase division multiplexing technique, which includes a phase-shifting circuit to enable multiple sensing coils to operate at different initial phases. The signals from the sensing coils are combined into a single output without loss of information. In a four-channel wear debris sensor, multiple signals can be simultaneously detected with a detection limit of 33 µm for ferrous wear debris. Additionally, Wu et al. [104] also proposed a multi-channel abrasive particle sensor system based on time division multiplexing, which can detect ten channels of abrasive particles simultaneously, as shown in Figure 8d. This system significantly improves oil throughput and avoids the crosstalk effect and burst noise observed in previous studies.

Despite these limitations, inductance-based counters continue to be an important tool in the field of particle counting and detection. Researchers are constantly exploring new designs and techniques to improve the sensitivity and versatility of inductance-based counters. For example, the use of hybrid sensors that combine multiple detection principles, such as inductance and capacitance, can improve the accuracy and range of particle detection [105]. Moreover, the integration of microfluidic channels with inductance-based counters can enhance the ability to detect and manipulate particles in complex environments. As the demand for precise and reliable particle counting and detection increases in various fields, including biomedical and environmental applications, the development of innovative inductance-based counters will continue to play a critical role in advancing the field.

### 3.4. Impedance-Based Counter 

In many cases, resistance, capacitance, and inductance all exist between two electrodes near the aperture. So, when particles pass through the aperture, impedance (which includes resistance, capacitance, and inductance) changes and contributes to the current pulse for detection. This time, the impedance-based counter would be the appropriate choice, and the principle of it is similar to the counters above. The schematic of the impedance-based counter is shown in Figure 9a [111].

The sensitivity of the impedance-based counter is a crucial factor in particle detection. One popular method to enhance sensitivity is to reduce the effective aperture by using non-conductive fluid to focus the sample. This technique has been demonstrated to provide more than a fivefold increase in the impedance signal by Winkler et al. [112]. Another method for increasing sensitivity is to integrate a resonator into the system. Haandbæk et al. [113] developed an impedance-based cytometer with a series resonant circuit. The resonant circuit comprised a discrete inductor connected in series with the capacitance and resistance of two opposing electrodes within a microfluidic channel. The enhanced sensitivity stemmed from the perfect impedance match between the inductor and the microfluidic channel when stimulated at the resonance frequency. Analogous to a balanced scale, where a slight weight change can tip the balance, even a minor impedance variation can lead to a substantial phase shift in the current passing through the resonator, resulting in heightened sensitivity. Liu et al. [114] proposed a parallel resonance circuit optimization technique by adjusting capacitance, which was used to improve the sensitivity of detecting wear debris in lubrication oil, as shown in Figure 9b. Based on the experimental data, a functional relationship between the parallel capacitance and the relative impedance variation is established. Thus, the optimal capacitance is found, which can maximize the absolute value of the relative impedance variation, as a way to improve the sensitivity.

The design and improvement of electrodes also contribute to the enhancement of sensitivity. Printed circuit boards (PCBs) are commonly used as electrodes due to their advantages, such as being reusable, cost-effective, and portable [115,116,117]. Later, Guler et al. [118] proposed a simple method for fabricating three-dimensional (3D) microelectrodes for impedance-based counters. The microelectrodes were etched from a microwire placed across a microchannel, and the electrode gap was precisely controlled using a hydrodynamically focused microfluidic device. With 3D microelectrodes, the detection of 6 μm diameter polystyrene beads was achieved. Tang et al. [119] have also presented liquid electrodes for the impedance counter, which were constructed by inserting Ag/AgCl wires into electrode chambers filled with highly conductive electrolyte solutions. This approach simplified the fabrication process and achieved high detection sensitivity.

Additionally, Sobahi et al. [120] developed a label-free and low-cost cell counting method that enables multi-channel, high-throughput cell counting using a single impedance analyzer assay, as shown in Figure 9c. The method utilizes only a pair of step-shaped impedance electrodes to measure the flow of isolated and sorted cells through different outlets. A pair of electrodes with five steps is integrated into five exit channels with different electrode-to-electrode distances, resulting in different electric field strengths between the electrodes in each channel when an AC voltage is applied. The differences in electric field strength produce different impedance signal signatures as cells flow through each electrode pair in each exit channel, allowing identification of the specific channel through which the cells pass.

In addition to the applications discussed in the introduction, impedance-based counters have found use in various other areas, including stem cell differentiation state identification [121], particle size measurement [111], droplet counting [122], and particle position detection [123]. Overall, impedance counters are attractive due to their label-free operation, simplicity, and cost-effectiveness. However, it is important to note that the electric field generated by the electrodes may cause damage to certain biological particles, which limits their application in some biological fields.

**Figure 9 micromachines-14-01722-f009:**
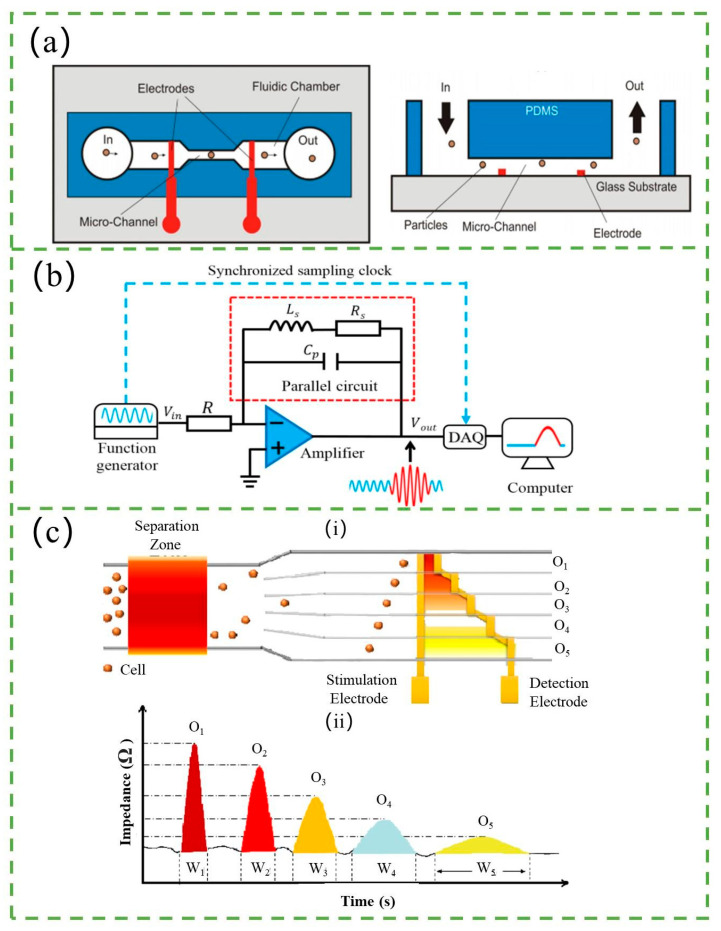
(**a**) Schematic diagram of the impedance-based counter and its equivalent electric circuit. Reproduced with permission from Ref. [111]. Copyright © 2023 IOP Publishing Ltd. (**b**) Schematic diagram of the experimental setup for signal detection using the parallel LC resonance circuit. Reproduced with permission from Ref. [114]. CC BY license. (**c**) Principle of the multi-outlet cell counting method employing a single pair of impedance electrodes. (**i**) Conceptual diagram illustrating the varying electrode-to-electrode distances for each outlet channel. Smaller gaps result in higher electric field strength compared to larger gaps, resulting in different electric field exposure for cells passing through different outlet channels (O1 to O5). (**ii**) Illustration of predicted impedance signal peak height and width variation when cells pass through different outlets, indicating expected differences in both peak height and width. Reproduced with permission from Ref. [120]. Copyright © 2023 Elsevier B.V.

## 4. Other Counters

### 4.1. Photoacoustic-Based Counter

Photoacoustic (PA) flow cytometry is a non-invasive and label-free technique that enables particle detection and counting based on acoustic waves generated by the photoacoustic effect. When a pulsed laser beam is absorbed by the particle, it generates a thermal expansion, which in turn generates acoustic waves that can be detected by an ultrasound transducer. The amplitude of the acoustic signal is proportional to the amount of absorbed light, and therefore to the particle’s size and optical properties. In PA flow cytometry, the particles are hydrodynamically focused into a single stream and illuminated by a laser beam. The resulting acoustic waves are detected and analyzed to obtain information about the particles, such as size, shape, and composition. PA flow cytometry has been used for the detection and analysis of various particles, including red blood cells, bacteria, and cancer cells. It offers high sensitivity, real-time detection, and the potential for multiplexed analysis of multiple parameters. 

Song et al. [124] demonstrated the use of the opto-acoustic fluidic microscopy for label-free detection of droplets and cells in microfluidic networks, as shown in Figure 10a. They were able to monitor droplet formation kinetics and mixing processes within droplets with high spatial and temporal resolution. Through their study on the relationship between the size of the photoacoustic signal and the concentration of molecular species in the microfluid, they applied photoacoustic imaging to count red blood cells encapsulated in water droplets of different sizes The number of red blood cells detected was found to increase in proportion to the volume of the droplets. They also developed a transmission-mode photoacoustic microscopy system with improved spatio-temporal resolution and a threefold increase in the frame rate compared to previous work, up to 2.5 kHz, to enhance analytical resolution and throughput [125]. The development of a label-free microfluidic acoustic flow cytometer (AFC) by NIGnyawali et al. [126] is a notable achievement in the field, as shown in Figure 10b. The AFC employs interleaved detection of ultrasound backscatter and photoacoustic waves to detect individual cells and particles within the microfluidic channel. The ultrasound utilized operates at a center frequency of 375 MHz (with a wavelength of 4 μm), while a nano-pulsed laser is employed for detection. The counting of red and white blood cells, as well as polystyrene particles, was performed using the AFC with blood samples of varying colors. The obtained results were closely compared to data from fluorescence-activated cell sorting (FACS). This approach offers a label-free alternative to traditional cell counting methods and has potential applications in various biomedical fields.

Acoustic-based counting is a novel label-free and non-destructive technique for rapid and multi-parametric analysis of diverse populations of individual cells. This technique utilizes ultrasound backscatter, light absorption, and physical properties as parameters for cell counting and sizing in biomedical and diagnostic applications [127]. However, this method has the disadvantage of complex equipment and a high cost. It should be noted that while acoustic-based counting has many advantages for cell counting and sizing, it also has some disadvantages. In addition to the complexity of equipment and high cost, the sensitivity and specificity of the acoustic-based counting technique may also be affected by factors such as the physical properties of the cells, the frequency and power of the ultrasound used, and the presence of background noise. Therefore, it is important to carefully consider the specific application and optimize the acoustic-based counting system accordingly to achieve accurate and reliable results.

**Figure 10 micromachines-14-01722-f010:**
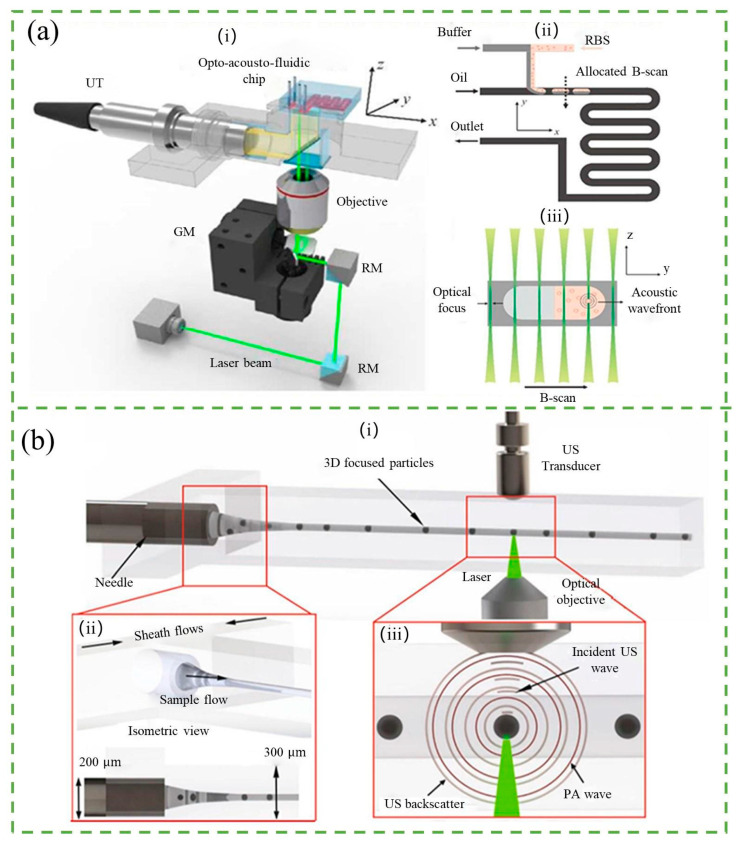
(**a**) (**i**) Schematic diagram of the opto-acousto-fluidic imaging module, where a pulse laser beam is directed through a reflective mirror (RM), galvanometer mirror (GM), and objective (4X) to illuminate the opto-acousto-fluidic chip orthogonally. (**ii**) Microfluidic network layout featuring a T-junction for droplet generation and a serpentine channel for passive mixing. (**iii**) Probe layout for optoacoustic signal generation and detection. Reproduced with permission from Ref. [124]. Copyright © 2023, Royal Society of Chemistry. (**b**) Conceptual schematic illustration of the acoustic flow cytometry system. (**i**) Overall design of the microfluidic device with collinearly aligned ultrasound (US) transducer and laser focusing optical objective. (**ii**) Hydrodynamic 3D flow focusing of the sample flow within the microfluidic device. (**iii**) Magnified view of the collinearly aligned transducer and laser beam, where the incident US wave and laser beam interact with a particle at the interrogation zone, producing both US backscatter and photoacoustic (PA) waves. Reproduced with permission from Ref. [126]. CC BY license.

### 4.2. Magnetic-Based Counter

Under certain conditions, a significant change in electrical resistance can be observed in adjacent ferromagnetic layers, known as giant magnetoresistance (GMR). When the magnetization of the layers is in parallel alignment, the overall resistance is relatively low, while it is relatively high for antiparallel alignment. By applying an external magnetic field, the direction of magnetization can be controlled, and this property can be utilized to develop GMR sensors. Integration of these sensors onto microfluidic chips enables the development of a magnetic-based counter. As magnetic particles pass through the microchannel and interact with the GMR sensor, their magnetization direction changes, which can be detected and used to count the particles, as shown in Figure 11a.

There are mainly two types of GMR sensors integrating with the microchip: spin-valve sensors (SVs) and tunneling magnetoresistive sensors (TMRs). Loureiro et al. [129,130] presented a magnetic-based counter that integrated SVs with the microfluidic channel. In their experiment, Kg1-a cells magnetically labeled with 50 nm CD34 microbeads (Milteny) and flowing at speeds of around 1 cm s^−1^ through a 150 µm wide, 14 µm high microchannel produced bipolar signals with an average amplitude of 10–20 mV corresponding to cell events. After that, Fernandes and Duarte et al. [127,131] presented an example for the validation of the platform’s integration with SVs that identified and quantified Streptococcus agalactiae in milk. Recently, Kokkinis et al. [132] demonstrated cancer cells labeled by MNPs and detected by SVs on the microfluidic chip. In contrast to SVs, TMRs use a thin insulating layer between the ferromagnetic layers, which results in a higher magnetoresistance effect. Gooneratne et al. [133] reported a microchip based on TMR for counting superparamagnetic beads (SPBs), which are magnetic particles with diameters typically in the range of 10–100 nm. They also utilized a unique magnetic actuator (MA) for the manipulation of SPBs. In their experiment, the SPBs flowed through the microfluidic channel and passed through a TMR sensor that detected their magnetic moment. The sensor was able to detect individual SPBs and provide a count of their number. 

Integrating magnetic biosensing and microfluidic systems onto a chip has become a growing trend in engineering research. Several research teams have focused on this and successfully developed fully functional magnetic flow cytometry systems [128]. For instance, Lee et al. [134] developed a magnetic flow cytometry system that consists of a microfluidic system and a GMR biosensor. By using hydrodynamic focusing techniques, clusters of different cell types are separated into individual cells and flow across a Wheatstone bridge consisting of four GMR biosensors. A high-speed camera captures the moment of cell flow across the GMR sensors, and the stray magnetic field/magnetic signal detected by the GMR is different due to the different endocytosis capabilities of the different cell types. The different cell types are then sorted and collected in different microfluidic channels and storages based on their response to magnetic forces. The system not only enables the counting of various cells but also offers detection and sorting functions.

The sensitivity of the magnetic counter is mainly determined by the GMR sensor, therefore, developing high-sensitivity GMR sensors is crucial to improve the sensitivity of the magnetic counter. Additionally, the strength of the magnetic field generated by the particle or label is also a critical factor. Magnetic sensors are highly versatile and can be easily integrated at a low cost, enabling the construction of complete signal processing systems on the same chip using related technologies [135,136]. Overall, the combination of magnetic biosensing and microfluidic systems on a chip has great potential for a wide range of applications in biomedical and environmental fields.

### 4.3. Thermal-Based Counter 

During the flow of a fluid through a microchannel, heat transfer is influenced by various factors such as the thermal conductivity of the fluid, the thermal resistance of the structure, and the flow profile. However, with a stable flow, the device reaches an equilibrium state after a certain time. When a particle enters the microchannel, the equilibrium is disturbed and the change in thermal conductivity causes a change in temperature, which can be measured using a thermal sensor [137]. This allows for the detection of particles within the flow.

A thermal-based counter was first used to detect droplets on the microfluidic chip. Yi et al. [138] proposed a novel real-time method for droplet detection and determination of protein concentration using the 3ω technique. AC power with frequency ω was applied to the metal heater, resulting in heating of the sample at frequency 2ω, and the detected signal was expressed at a frequency of 3ω. By monitoring the thermal response of both droplets and the carrying flow, water droplets within an oleic acid carrying flow were successfully detected. Later, Vutha et al. [139] proposed a microfluidic device for thermal particle detection. Their experimental setup is illustrated in Figure 12. The PDMS with the microchannel was placed on the silicon substrate containing the resistance temperature detector (RTD). Their counter was able to detect particles with diameters of 90 and 200 µm, with multiple particles counted in series to demonstrate its utility. The high sensitivity of the Vox micro-thermistor for the detection of biomolecules based on enzymatic reactions makes it a promising tool in biomedical research. Inomata et al. [140,141] developed a highly sensitive thermal measurement device based on a vanadium oxide (VO_x_) micro-thermistor for the detection of biomolecules based on enzymatic reactions. The temperature dependence of the resistance of the thermistor is due to the heat generated by enzymatic reactions, allowing for the detection of biomolecules. The device consists of a microfluidic channel and chamber, and a VO_x_ thermistor on a suspended Si_3_N_4_ membrane for thermal insulation. In addition to glucose and cholesterol, this device has the potential to detect a wide range of other biomolecules. Inomata et al. [142] also presented a novel sensor consisting of two microfluidic reservoirs, a thermal bench, and three electrodes. PEG-NaOH and iodine-based aqueous solutions were used as positive and negative ionic liquids, respectively. The output voltage of the sensor increases linearly with the input temperature. The device has a Seebeck coefficient of 10.6 mV/K and a temperature resolution of 8.94 mK, making it a promising candidate for temperature-based detection applications.

The thermal counter provides a new method for counting without optical or electric fields. It has many advantages, such as label-free, in situ, real-time analysis, and so on. However, there are also some disadvantages. The sensitivity of the thermal counter is determined by the thermal sensor, and it is difficult to count small particles such as nanoparticles. Additionally, the thermal counter is affected by changes in the flow rate and temperature, which may result in fluctuations in the measured signals. Meanwhile, the thermal counter requires precise control of the temperature, which may limit its application in some fields. Despite these limitations, the thermal counter shows great potential for various applications, such as droplet detection, particle counting, and biomolecule detection. Further improvements in the design and development of thermal sensors may enhance the sensitivity and accuracy of thermal counting devices, making them more widely applicable in research and industry.

## 5. Future Directions and Challenges

Significant progress has been made in developing various microfluidic systems. The main types of microfluidic counters and their characteristics are summarized by Table 1. Additionally, microfluidic technology’s enhancement of experimental methods, cost reduction, and simplification has drawn broad attention in biotechnology. Microfluidic devices have diverse applications in life sciences, including real-time healthcare, precision and personalized medicine, regenerative medicine, prognosis, diagnostics, and treatment of tumor-related and non-tumor-related ailments. For instance, silver nanoparticles have gained notice due to their lack of microbial or viral resistance, making them useful for infection prevention and control [143]. Utilizing these traits, AV Blinov et al. have synthesized silver nanoparticles and oxidized variants, exploring their potential for suture coating components [144].

Micro-Electro-Mechanical Systems (MEMS) are intricate miniaturized devices often produced using microfabrication methods. They cleverly combine mechanical and electrical components to perform tasks akin to those accomplished by larger systems [145]. MEMS offer benefits like compact size, seamless integration, low weight, minimal power usage, and high resonant frequencies [146]. These attributes have led to growing interest in their use within the biomedical realm. Progress in computational technologies has facilitated the integration of microfluidic approaches into advanced MEMS device design, yielding advantages such as reduced energy consumption, limited reagent consumption, and enhanced detection sensitivity [147].

In biomedicine, MEMS technology has made notable strides and is recognized as BioMEMS. These devices are designed and manufactured for real-time disease diagnosis, biosensors, drug delivery systems, and surgical tools [148,149]. MEMS technology is widely utilized as a platform for producing enhanced and uniform nanoparticles. Additionally, wearable MEMS devices have become vital for individuals with chronic conditions, enabling remote monitoring of crucial signs like blood pressure, intracranial pressure, blood glucose levels, heart and respiratory rates, body temperature, and oxygen saturation [150].

**Table 1 micromachines-14-01722-t001:** Summary of the main types of microfluidic counters and their characteristics.

Classification	Method	Limit of Detection	Integration Difficulty	Instrument Price	Advantages	Disadvantages
Optical-Based Counters	Fluorescence Counter	30 cfu/mL (*Escherichia coli* O157:H7)	Medium	Moderately high	High sensitivity, multiple labeling	Complex equipment
	Optical Absorbed Counter	400 nM (Rhodamine 6G concentration)	Low	Low	No labeling, easy to operate	Lower flux, low light transmission, poor effect
	Scattering Counter	1 μm (Particle size)	High	High	Label-free, differentiates between different particles	Complex equipment, difficult to integrate
	Refractive Index Counter	$4.3\times 10^{7}\;\mathrm{mV}/RIU$(Refractive index change)	Low	Medium	Label-free, can monitor cell status	Mainly used for analytical testing, not suitable for opaque particles
Electrical-Based Counters	Resistance-Based Counter	1 μm (Particle size)	Low	Low	Simple, label-free, low cost	Only suitable for conductive solutions, limited flux
	Capacitance-Based Counter	95 μm (Bubble diameter)	High	Medium	Can be used in non-conductive solutions	Complicated electrode production, low sensitivity
	Inductance-Based Counter	11 μm (Abrasive particle diameter)	Low	Low	Simple, low cost	Only detect metal abrasion particles, low throughput
	Impedance-Based Counter	6 μm (Bollstein microsphere diameter)	Low	Low	Simple, low cost	Electric field may damage biological particles
Other Counters	Photoacoustic-Based Counter	No clear data	High	High	Label-free, real-time detection	Complex equipment, high cost
	Magnetic-Based Counter	20 μm (Iron particle diameter)	Low	Low	Integrable, Low Cost	Only magnetic particles can be detected
	Thermal-Based Counter	90 μm (Particle diameter)	Medium	Medium	Label-free, in situ detection	Low sensitivity to small particles, susceptible to flow velocity and temperature

Microfluidic devices face significant challenges in data analysis, emphasizing the integration of artificial intelligence for enhanced analytics. Material limitations, particularly with widely-used materials like PDMS, necessitate exploration of alternatives such as SEBS. Additionally, the manufacturing of microfluidic devices, especially MEMS devices, is beset by issues like fragility and contamination, requiring innovative solutions to improve their production processes. These multifaceted challenges collectively underscore the need for continuous advancement in microfluidic technology.

### 5.1. Data Analysis Capabilities

As controlled reaction chambers, high-throughput arrays, and positioning systems advance, microfluidics has amassed substantial data. Yet, not all data are readily analyzable. Thus, leveraging potent artificial intelligence for analysis becomes imperative, tightly fusing it with microfluidic devices to synergistically augment their capabilities. With the advent of machine learning, numerous advantages for particle counting based on microfluidic counters have emerged. Recent investigations have been focused on the development of machine and deep learning algorithms aimed at training models for the assessment and categorization of imaging flow cytometry images, thereby enhancing analytical workflows [151,152,153]. Machine learning offers the potential for optimizing parameters and models, significantly reducing processing time [154]. Moreover, contemporary research in the realm of flow cytometry encompasses cell classification, cell isolation, and their synergistic integration, benefiting from the integration of artificial intelligence techniques and microfluidic methodologies [155].

### 5.2. Material

Polydimethylsiloxane (PDMS) is highly favored for microfluidic device manufacturing due to its rapid prototyping, user-friendliness, and biocompatibility advantages [156]. However, PDMS does have limitations. For instance, it is prone to absorbing small hydrophobic molecules, constraining manufacturing processes, parameters, and chip dimensions. Thus, the pursuit of alternative materials with superior performance is necessary. Recently, materials like styrene-butadiene-styrene (SEBS) gained wide application for their balanced mechanical properties, processability, and recyclability [157,158,159].

MEMS devices confront challenges in their overreliance on silicon and derivatives as exclusive materials [160]. It prompts materials science to boldly explore and develop to meet new material needs in microdevices.

### 5.3. Equipment Manufacturing

Numerous microfluidic devices encounter manufacturing challenges. MEMS devices, owing to their small size, fragility, and susceptibility to external influences, can experience problems like cracking, bending, or misalignment of moving parts [161]. Moreover, due to their complex mechanical structures, these devices are prone to issues arising from particle contamination, fatigue, fractures, friction, or surface adhesion.

## 6. Conclusions

This paper provides a comprehensive review of recent developments in microfluidic counters, including their detection principles, latest research findings, and advantages and disadvantages. Many papers also discuss strategies to enhance the sensitivity and output of these counters. In future research, how to consider from multiple perspectives, more comprehensively weigh sensitivity and throughput, thereby straddling performance metrics sans undue compromise will be a challenge. In addition, there has been increasing interest in miniaturizing [93] and making portable or wearable microfluidic counters [11,162]. Recent research has also focused on integrating two or more types of counters onto a single chip, which is likely to be a major direction for future research in this field [163,164].

## Figures and Tables

**Figure 1 micromachines-14-01722-f001:**
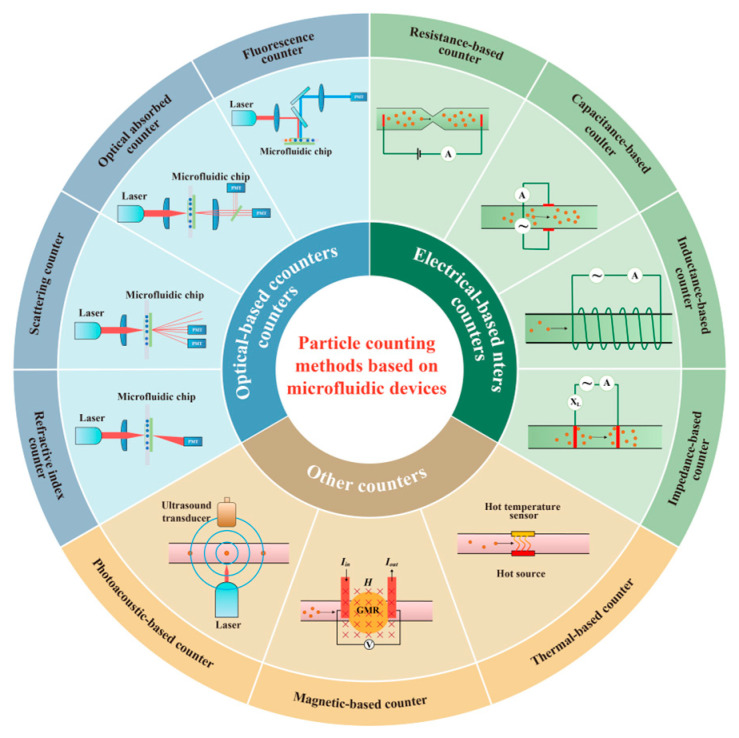
Diagram of particle counting methods based on microfluidic devices.

**Figure 3 micromachines-14-01722-f003:**
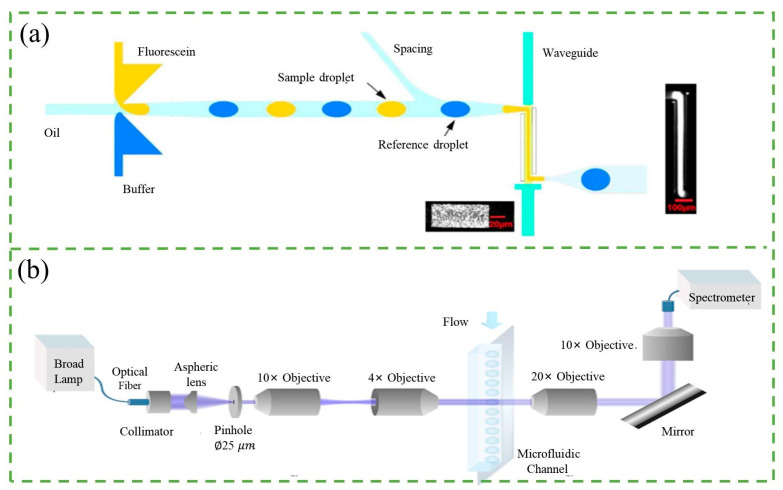
(**a**) Absorption spectrometry-based droplet detection. Schematic of an optofluidic device with a Z-shaped detection path and two liquid-core (PDMS) waveguides. Reproduced with permission from Ref. [40]. Copyright © 2023 American Chemical Society. (**b**) Schematic diagram of the optical absorption counter. A high-intensity broad-band light source is collimated and directed through a 25 μm pinhole, resulting in a 75 μm diameter circular detection area on the microfluidic channel. Each droplet is positioned within this detection area. The light exiting the channel is collected and focused onto an optical fiber connected to a fast spectrometer. Spectra were recorded at 4500 Hz with a 50 μs acquisition time. Reproduced with permission from Ref. [41]. Copyright © 2023 American Chemical Society.

**Figure 4 micromachines-14-01722-f004:**
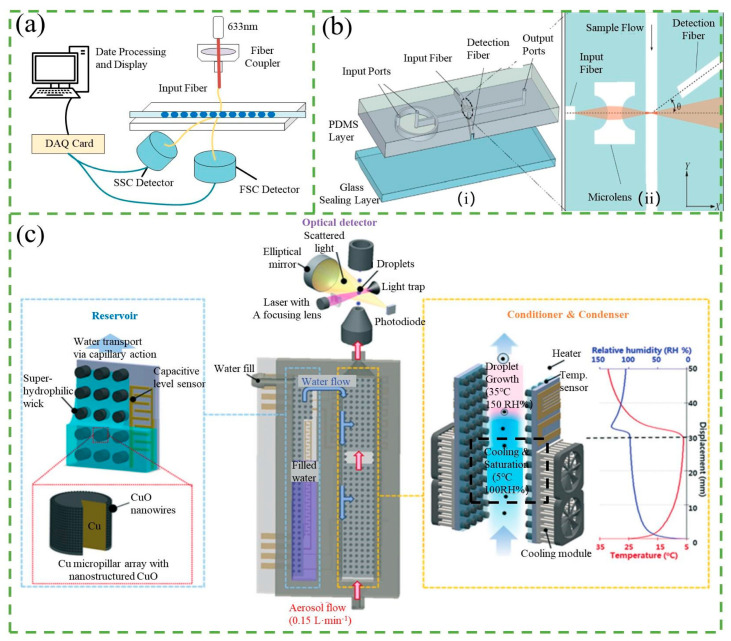
(**a**) Schematic diagram of the scattering counter capable of detecting forward scatter (FSC) or side scatter (SSC) for particle counting. (**b**) (**i**) A 3D structural depiction of the microfluidic cytometer. (**ii**) Detailed schematic of the integrated optical systems. Reproduced with permission from Ref. [54]. Copyright © 2023 AIP Publishing. (**c**) Schematic diagram of the detection system utilizing condensation nucleation light scattering technology. Reproduced with permission from Ref. [55]. Copyright © 2023 Royal Society of Chemistry.

**Figure 5 micromachines-14-01722-f005:**
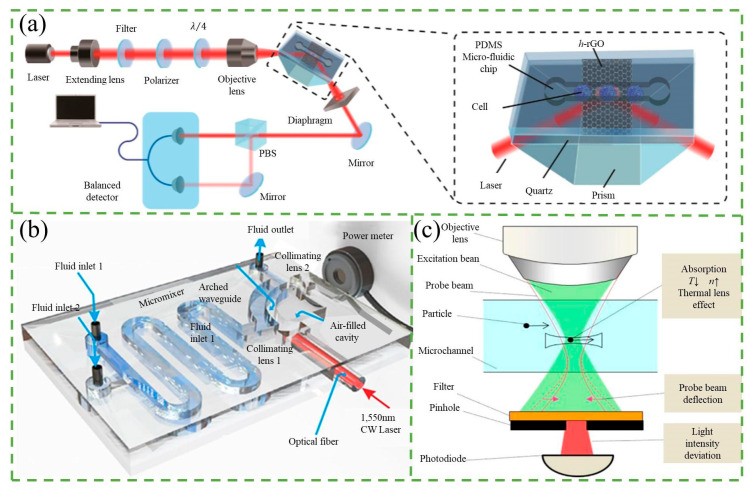
(**a**) Schematic diagram of the graphene-based optical single-cell sensor, featuring a PDMS microfluidic chip/h-rGO/quartz sandwich structure on a prism. Reproduced with permission from Ref. [66]. Copyright © 2023 American Chemical Society. (**b**) Schematic diagram of the arched optofluidic waveguide refractive index (RI) sensor. Reproduced with permission from Ref. [67]. Copyright © 2023 The Optical Society. (**c**) Principle of thermal lens nanoparticle detection in a microchannel. Reproduced with permission from Ref. [68]. Copyright © 2023 Elsevier B.V.

**Figure 6 micromachines-14-01722-f006:**
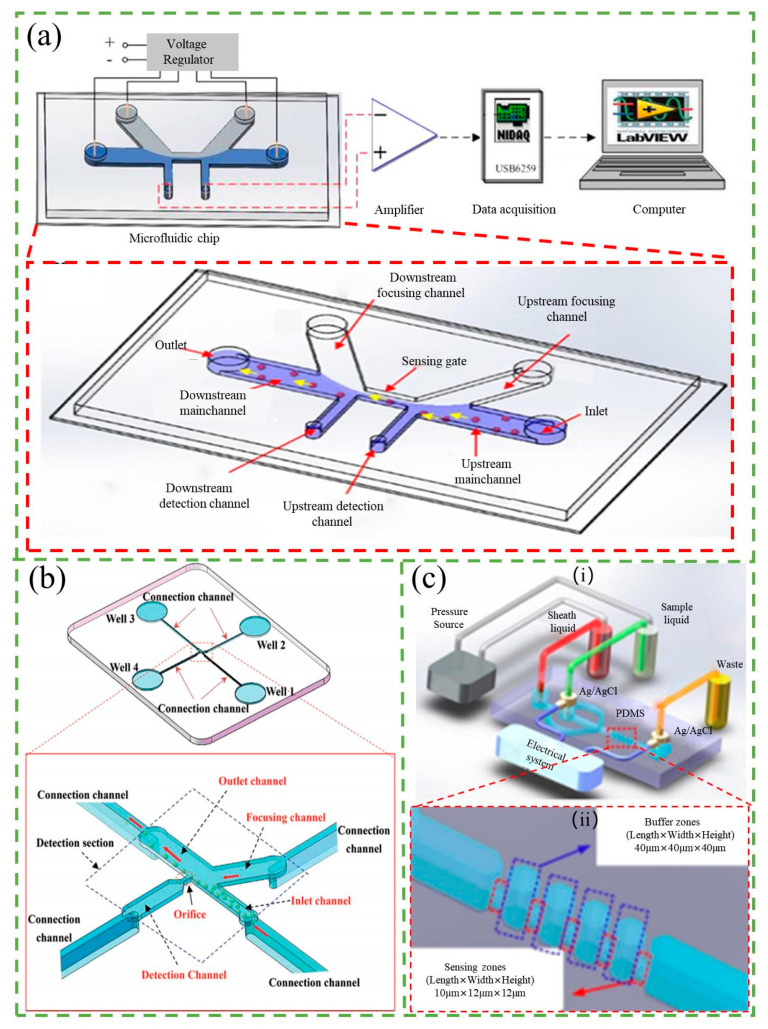
(**a**) Schematic diagram of the system setup and microfluidic chip structure utilizing a novel electrokinetic focusing method. Reproduced with permission from Ref. [78]. Copyright © 2023 Springer-Verlag Berlin Heidelberg. (**b**) Schematic diagram of the side-orifice-based resistive pulse sensor (RPS). Reproduced with permission from Ref. [79]. Copyright © 2023 Royal Society of Chemistry. (**c**) (**i**) Principle of multiple sensing pores for enhanced size discrimination. Ag/AgCl electrodes are inserted into the tubes containing the sample liquid and waste, allowing voltage application and current measurement. Sequential traversal of sensing pores by particles generates a signal with multiple drop pulses. (**ii**) Enlarged schematic diagram of the sensing zones and buffer zones. Reproduced with permission from Ref. [80]. Copyright © 2023 Elsevier B.V.

**Figure 8 micromachines-14-01722-f008:**
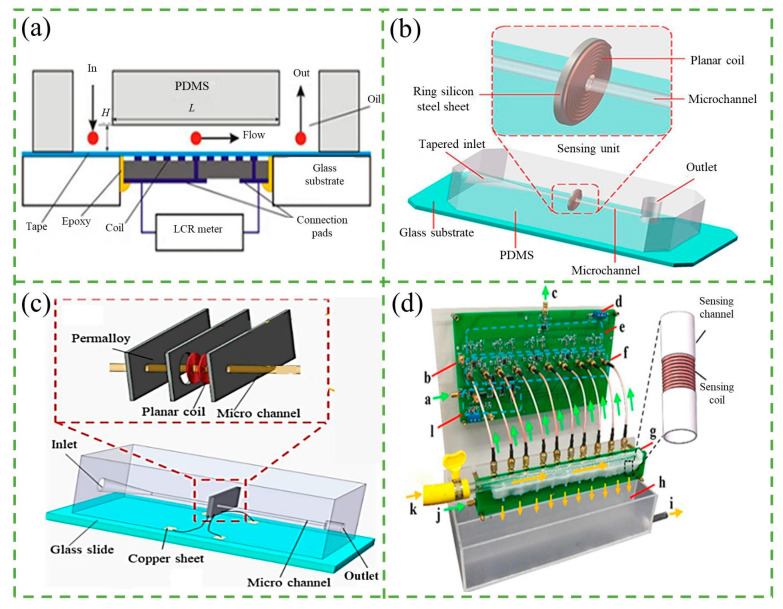
(**a**) Schematic diagram of the microfluidic inductance counter. Reproduced with permission from Ref. [101]. Copyright © 2023, Springer-Verlag. (**b**) Schematic diagram of the integrated sensor featuring a ring-shaped silicon steel sheet. Reproduced with permission from Ref. [102]. Copyright © 2023 Elsevier Ltd. (**c**) Schematic diagram of the oil debris sensor: (i) Three-dimensional structure of the sensor. (ii) Enlarged schematic diagram of the inductance unit. Reproduced with permission from Ref. [103]. CC BY license. (**d**) Measurement setup of a ten-channel system with the following components: (a) Signal input for generating ten square waves. (b) Test port. (c) Signal output. (d) Power supply (DC, ±5 V). (e) Units 3 and 4. (f) Unit 2, the summing circuit of two signals. (g) Unit 5, the voltage follower, and unit 6, ten sensor circuits. (h) Oil reservoir. (i) Oil outlet. (j) Signal input for exciting ten sensing coils. (k) Oil inlet. (l) Square wave generator. Reproduced with permission from Ref. [104]. Copyright © 2023 IEEE.

**Figure 11 micromachines-14-01722-f011:**
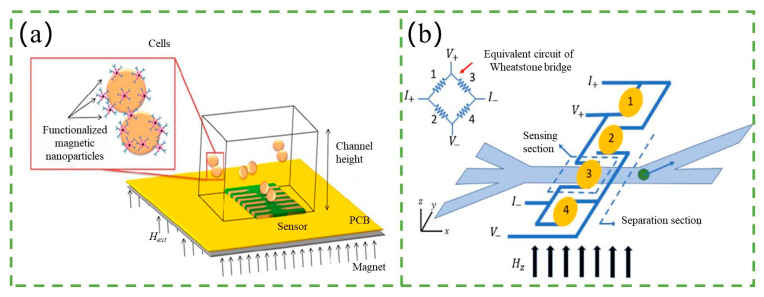
(**a**) Schematic diagram of a magnetic detection device for cell identification and quantification. Reproduced with permission from Ref. [127]. CC BY license. (**b**) Microfluidic system integrated with magnetic sensors for cell counting and sorting. A cell labeled with magnetic nanoparticles can be detected by a Wheatstone bridge consisting of four GMR sensors. Reproduced with permission from Ref. [128]. Copyright © 2023 World Scientific Publishing Company.

**Figure 12 micromachines-14-01722-f012:**
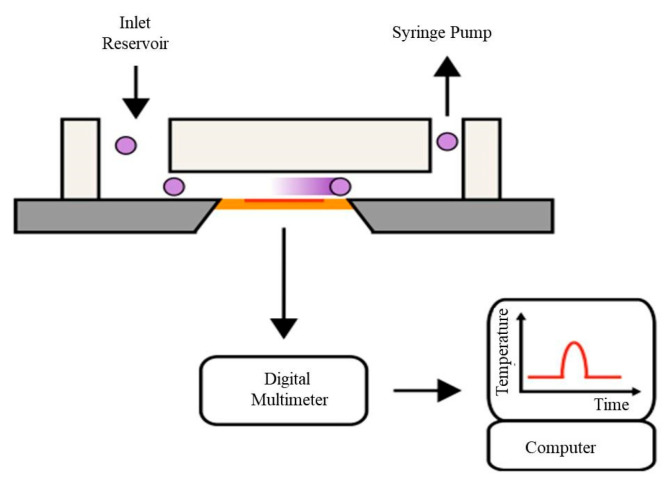
Schematic diagram microfluidic device for thermal particle detection. Reproduced with permission from Ref. [139]. Copyright © 2023, Springer-Verlag Berlin Heidelberg.

## Data Availability

No new data were created or analyzed in this study.

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
