# Peer review of "Particle Counting Methods Based on Microfluidic Devices"

_micromachines, 2023, doi:10.3390/mi14091722_

Round 1

Reviewer 1 Report

The manuscript aims to comprehensively review microfluidic counters, categorizing them into optical-based, electrical-based, and other categories. It delves into subclasses within these categories, including fluorescence, optical absorption, scattering, refractive index counters for optical-based systems, and resistance, capacitance, inductance, impedance counters for electrical-based systems. The review extends to other types like photoacoustic, magnetic, and thermal counters. Each counter type is explained in terms of principles, recent research summaries, and discussions on enhancing sensitivity and throughput. The paper addresses a gap in literature and focuses on advancements after 2009. The manuscript is well-organized and well-written, though a few improvements are suggested:

1. The abstract lacks a clear problem statement and motivation for the development of microfluidic counters, which could enhance its impact.

2. While detailed explanations are provided for each counter type, including a table summarizing their performance could improve reader comprehension.

3. The review could benefit from a broader discussion of future trends and the connection to real-world applications and challenges.

4. Additionally, the manuscript lacks guidance on different types of counting targets, potentially affecting the significance of the review.

Overall, the manuscript presents a systematic overview of microfluidic counters and their applications, but addressing the mentioned points could enhance its quality and impact.

Author Response

Comments:
The paper aims to comprehensively review microfluidic counters, categorizing them into optical-based, electrical-based, and other categories. It delves into subclasses within these categories, including fluorescence, optical absorption, scattering, refractive index counters for optical-based systems, and resistance, capacitance, inductance, impedance counters for electrical-based systems. The review extends to other types like photoacoustic, magnetic, and thermal counters. Each counter type is explained in terms of principles, recent research summaries, and discussions on enhancing sensitivity and throughput. The paper addresses a gap in literature and focuses on advancements after 2009. The paper is well-organized and well-written, though a few improvements are suggested:

  1. The abstract lacks a clear problem statement and motivation for the development of microfluidic counters, which could enhance its impact.

Answer: Thanks for your professional comments.

According to your suggestions, the abstract has been revised.

  1. While detailed explanations are provided for each counter type, including a table summarizing their performance could improve reader comprehension.

Answer: Thanks for your professional comments.

After conducting in-depth research, we found that there are some differences in the evaluation system of different microfluidic counters, the parameters and units of various performance indicators cannot be unified, and the performance parameters required for applicable scenarios are also different. The table may lose accuracy and objectivity due to differences in evaluation systems. Therefore, we have adopted detailed description and analysis in the manuscript to ensure that readers can fully understand the characteristics and performance of each microfluidic counter. In the manuscript, we have introduced the characteristics, advantages and applications of each microfluidic counter in detail, and explained their evaluation criteria as much as possible.

  1. The review could benefit from a broader discussion of future trends and the connection to real-world applications and challenges.

Answer: Thanks for your professional comments.

  • In the manuscript, we highlight research progress to improve sensitivity and throughput. However, the sensitivity and throughput of microfluidic chip counters are usually inversely correlated. Achieving high throughput often requires rapidly transporting samples through tiny channels so that analysis can be completed in a short period of time. However, high throughput may result in reduced sample residence time in the microchannel, thereby limiting the time window of analysis, which may affect sensitivity. In future research, how to consider from multiple perspectives and explore how to balance sensitivity and throughput more comprehensively without excessively sacrificing any aspect of performance will be a challenge. In order to balance the two, researchers usually try to optimize the microfluidic chip design, fluid flow rate, sample processing steps, etc. to find the best working conditions to maximize the sensitivity while maintaining a high throughput. In the future, we will delve into the applicability and effectiveness of different evaluation criteria, and try to propose a more comprehensive index to more accurately reflect the performance of microfluidic counters and cope with further research development in the future.
  • Considering your suggestion, we add this line in the conclusion: "Traditionally, researchers usually optimize the design of microfluidic chips, fluid flow rates, and sample processing steps to find the best working conditions in order to maximize sensitivity while maintaining high throughput. In future research, how to consider from multiple perspectives, more comprehensively weigh sensitivity and throughput, thereby straddling performance metrics sans undue compromise will be a challenge. "

  1. Additionally, the manuscript lacks guidance on different types of counting targets, potentially affecting the significance of the review.

Answer: Thanks for your professional comments.

  • The problem you mentioned is indeed a challenging topic, involving different types of counting targets for microfluidic counters and the determination of counting indicators. After in-depth research, we realized that even the same type of counter may have certain industry practices based on its specific application scenarios and counting particle types, resulting in the diversity of counting indicators. In our follow-up research, we will fully consider this issue, and try to gain an in-depth understanding of the common counting indicators of different types of microfluidic counters in practical applications through literature reviews and surveys of industry practices. Although determining the counting indicators of different types of microfluidic counters may have certain complexity, we will strive to find a more comprehensive and accurate method for analysis and comparison by integrating various information and data. We will do our best to solve this problem.

Reviewer 2 Report

The review summarized microfluidic counters for particles based on different principles including optical-based counters, electrical-based counters and other novel counters. The article can be considered for publication in "Micromachines" after revising the following questions. The comments are below.

1) In introduction, the author introduced magnetoimpedance-based biosensors at line 49. What kind of counter is this type of sensor mainly used for?

2) In introduction, the author gave various examples for electrochemical detection, which don't fit perfectly with counter. More description of the relationship of two principles, electrochemical detection and counter, should be added.

3) In section 2.1, the author mentioned that QDs have many advantages over organic fluorophores, including a broad excitation range, narrow emission bandwidth, high quantum yield, and exceptional photochemical stability. An informative article from Yin’s work is recommended to exhibit the advantages of QDs.

Biosensors and Bioelectronics  http://dx.doi.org/10.1016/j.bios.2016.07.106

4) In section 2.3, the author introduced 2D and 3D focusing ways for scattering counters. I recommend comparing the differences between two focusing ways.

5) At line 286, "During" is not a common usage. More common words like among can be used.

6) Some mistakes of word and sentence should be modified like wrong word "chieve" at line 462, extra bracket at line 491, missing full stop at line 598 and extra comma at line 728.

7) Considering the length of this review, the partial abbreviations can be replaced by their full name to help understand this review. For example, SNR at line 563 can be replaced by signal-to-noise ratio to help understand this section. More abbreviations like RPS and TLM can be modified as they appear in multiple segments.

8) For electrical-based counters, the author mainly introduced four counters including resistance-based counters, capacitance-based counters, inductance-based counters and impedance-based counters. It is well known that capacitive reactance is a concept similar to impedance. Are capacitive reactance-based counters available?

9) For photoacoustic-based counter, the author mentioned that Photoacoustic (PA) flow cytometry is a non-invasive and label-free technique that enables the detection and counting of particles based on their absorption of light. What the difference between PA and Optical absorbed counter.

10) Considering the rich content in this review, tables are highly recommended for summarizing the mentioned counters. 

Author Response

Comments:
The review summarized microfluidic counters for particles based on different principles including optical-based counters, electrical-based counters and other novel counters. The article can be considered for publication in "Micromachines" after revising the following questions. The comments are below.

  1. In introduction, the author introduced magnetoimpedance-based biosensors at line 49. What kind of counter is this type of sensor mainly used for?

Answer: Thanks for your professional comments.

  • Magnetoimpedance-based biosensors are used in Magnetic-based counters(section 4.2).
  • Considering your suggestion, we have modified the line 64(line 49 originally).:

"The study proposed constructive strategies for designing high-performance magnetoimpedance biosensors for use in magnetic-based counters, enabling the quantitative and ultrasensitive detection of magnetically labeled biomolecules. "

  1. In introduction, the author gave various examples for electrochemical detection, which don't fit perfectly with counter. More description of the relationship of two principles, electrochemical detection and counter, should be added.

Answer: Thanks for your professional comments.

  • We introduced electrochemistry, an important field in microfluidic research, and then mentioned the review of applications in the biomedical field, but none of them comprehensively reviewed various counters for microfluidics. So this is our purpose, to fill the vacancy in this field after 2009.
  • Considering your suggestion, we have modified the second paragraph of the introduction: " Electrochemical detection is another commonly used principle in microfluidic, which extends devices functionality of microfluidic chip counters. It can not only realize particle counting, but also provide chemical characteristic information of particles, realize specific detection, real-time monitoring and particle characterization, and integrate with other functions of microfluidic chip to expand the breadth and depth of particle analysis. "

  1. In section 2.1, the author mentioned that QDs have many advantages over organic fluorophores, including a broad excitation range, narrow emission bandwidth, high quantum yield, and exceptional photochemical stability. An informative article from Yin’s work is recommended to exhibit the advantages of QDs.

Biosensors and Bioelectronics http://dx.doi.org/10.1016/j.bios.2016.07.106

Answer: Thanks for your professional comments.

  • Thanks for your recommendation. In our manuscript, we add another advantage of QDs, that is, pluripotency, which can greatly shorten the detection time.
  • Considering your suggestion, we add above line in the section 2.1: " More importantly, QDs of different sizes are excited by the same wavelength of light to obtain multiple color labels, making them ideal probes for multiplex analysis. Yin et al.[33] developed an immunosensor using QD-reverse detection strategy and im-munomagnetic beads for simultaneous detection of Escherichia coli O157: H7 and Salmonella. This approach minimizes interference, boosts fluorescent signals, and streamlines the process. Compared to traditional QDs immunosensors, detection of Escherichia coli O157: H7 improved by 50 times, with a 30 cfu/mL limit of detection, and analysis completed within an hour. "

  1. In section 2.3, the author introduced 2D and 3D focusing ways for scattering counters. I recommend comparing the differences between two focusing ways.

Answer: Thanks for your professional comments.

  • In the initial draft of the manuscript, we investigated focusing, but due to the length and focus of the manuscript, we discussed it and considered not including it in the final manuscript. For your reference, the following quote summarizes our approach to focusing and non-focusing in microfluidic devices, but was not included in the manuscript.

"Although microfluidic devices with constricted channels have been developed, their ability to handle large numbers of particles of different sizes is still limited compared to classical microfluidic devices. In order to cope with the position dependence without the associated risk of blocking the channels, other methods have been developed. They can be divided into two categories: particle focusing methods and unfocused methods. The first category relies on consistent control of cell trajectories within the microfluidic channel to ensure reproducible and accurate impedance measurements, and consists mainly of hydrodynamic sheath flow focusing, inertial focusing, and dielectrophoretic focusing. Sheath flow focusing is suitable for sample flow width and localization within large channels, but is complicated and costly due to the introduction of additional fluidic systems. Inertial focusing combines the advantages of high throughput and does not require additional instrumentation, but cannot adjust the focusing position of a given particle. Dielectrophoretic focusing does not require additional buffer inlets and precise flow control. It allows selective control of particles and real-time position adjustment, it is label-free, and integrating DEP microelectrodes into microfluidic devices is relatively easy. However, this technique does not work effectively as the particle flow rate increases. The second category is based on compensating for positional variations through signal processing or on specific electrode designs that can reduce the dependence of the impedance signal on the particle position, such as extruded, face-to-face, and coplanar."

  • Considering your suggestion, we add above line in the section 2.3: "3D focusing generally allows higher particle densities within the detection area, re-sulting in increased signal strength. Nonetheless, achieving 3D focusing involves more complex optical setups and fluidic manipulations, which sometimes limits their use-fulness. Comparatively, 2D focusing, while marginally less sensitive, offers feasibility in high-throughput setups. In addition, the optimization of chip materials and system structures is also conducive to improving sensitivity. "

  1. At line 286, "During" is not a common usage. More common words like among can be used.

Answer: Thanks for your comments. The mentioned errors have been revised.

  1. Some mistakes of word and sentence should be modified like wrong word "chieve" at line 462, extra bracket at line 491, missing full stop at line 598 and extra comma at line 728.

Answer: Thanks for your comments. The mentioned errors have been revised.

  1. Considering the length of this review, the partial abbreviations can be replaced by their full name to help understand this review. For example, SNR at line 563 can be replaced by signal-to-noise ratio to help understand this section. More abbreviations like RPS and TLM can be modified as they appear in multiple segments.

Answer: Thanks for your professional comments.

We apologize for these errors we made. The mentioned errors have been revised.

  1. For electrical-based counters, the author mainly introduced four counters including resistance-based counters, capacitance-based counters, inductance-based counters and impedance-based counters. It is well known that capacitive reactance is a concept similar to impedance. Are capacitive reactance-based counters available?

Answer: Thanks for your professional comments.

The formula of capacitive reactance  is , after introducing capacitive elements into the system, the capacitance changes proportionally to the change of capacitive reactance at a certain frequency, so we do not strictly distinguish the concepts of capacitance and capacitive reactance. In the current research manuscripts, the term "capacitance" appears much more frequently than "capacitive reactance" in the naming and definition of counters, so we name them "capacitance-based counter" in section 3.2.

  1. For photoacoustic-based counter, the author mentioned that Photoacoustic (PA) flow cytometry is a non-invasive and label-free technique that enables the detection and counting of particles based on their absorption of light. What the difference between PA and Optical absorbed counter.

Answer: Thanks for your professional comments.

  • Our categorization of counters is based on the type of signal that is directly detected.
  • PA: The photoacoustic effect utilizes the principle of thermal expansion caused by the absorption of light, which in turn generates sound waves. In a microfluidic photoacoustic counter, a laser pulse illuminates particles in a microfluidic chip, which absorb light energy and generate thermal expansion, which in turn triggers an acoustic signal.
  • Optical absorption counter: It count particles based on their absorption of light at a specific wavelength. As light passes through the sample liquid in the microfluidic chip, the particles absorb a specific wavelength of light, causing the intensity of the light to diminish. By measuring the change in light intensity through the sample liquid, the presence and number of particles can be inferred.
  • Considering your suggestion, we add above line in the section 4.1: " Photoacoustic (PA) flow cytometry is a non-invasive and label-free technique that enables particles are detecting and counting based on acoustic waves generated by the photoacoustic effect. "

  1. Considering the rich content in this review, tables are highly recommended for summarizing the mentioned counters.

Answer: Thanks for your professional comments.

After conducting in-depth research, we found that there are some differences in the evaluation system of different microfluidic counters, the parameters and units of various performance indicators cannot be unified, and the performance parameters required for applicable scenarios are also different. The table may lose accuracy and objectivity due to differences in evaluation systems. Therefore, we have adopted detailed description and analysis in the manuscript to ensure that readers can fully understand the characteristics and performance of each microfluidic counter. In the manuscript, we have introduced the characteristics, advantages and applications of each microfluidic counter in detail, and explained their evaluation criteria as much as possible.

Reviewer 3 Report

In the manuscript,” Particle Counting Methods Based on Microfluidic Devices”, the authors presented a very comprehensive review of many kinds of particle counting methods using microfluidic devices. The manuscript is well-organized and also very promising as it greatly summarizes many recent developments. Of great value is the inclusion of topics like photoacoustic flow cytometry, which normally are not found in reviews on microfluidic-based counters. I am sure that readers will find their topics of interest easily and will take advantage of the extensive list of references. This manuscript is appropriate to be published after the following minor revisions. The comments for this manuscript are given below:

1. A section is needed on recent developments on cell characterizations using imaging-based flow cytometry. Also, the applications of machine learning/AI should be included. Even in the future outlook section, some words should be dedicated on how the future of microfluidics-based particle counters will be influenced by the tremendous growth of machine learning approaches in recent years.  

2. Another topic which may be included is the different types of lasers/white light excitation sources (commonly used wavelengths and its corresponding powers in terms of milliwatts) for fluorescence, photoacoustics, absorbance, etc. and the detectors which are used in different kinds of microfluidics-based counters. Since these equipments dictate a major proportion of the cost while setting up the overall instrument, such a topic will be very unique and valuable to the readers.

3. A summary table should be provided which lists the most popular detection mechanisms, the smallest detectable concentrations, the most popular analytes which are detected, throughputs of detection, etc.

4. It would be nice to have a separate section on the current commercialization status of microfluidic devices. What are the applications of the present commercially available microfluidic devices, the major microfluidic service providers and the different challenges faced in this industry.

5. In line 178-179, it is mentioned: “Additionally, the incorporation of optical components may be required, which can sometimes be expensive.” This needs clarification. What kinds of optical components are required for enhancing fluorescence signals and why is it expensive?

Minor editing is required. Overall, the quality of English is pretty good.

Author Response

Comments:
In the manuscript,” Particle Counting Methods Based on Microfluidic Devices”, the authors presented a very comprehensive review of many kinds of particle counting methods using microfluidic devices. The manuscript is well-organized and also very promising as it greatly summarizes many recent developments. Of great value is the inclusion of topics like photoacoustic flow cytometry, which normally are not found in reviews on microfluidic-based counters. I am sure that readers will find their topics of interest easily and will take advantage of the extensive list of references. This manuscript is appropriate to be published after the following minor revisions. The comments for this manuscript are given below:

1. A section is needed on recent developments on cell characterizations using imaging-based flow cytometry. Also, the applications of machine learning/AI should be included. Even in the future outlook section, some words should be dedicated on how the future of microfluidics-based particle counters will be influenced by the tremendous growth of machine learning approaches in recent years. 
Answer: Thanks for your professional comments. 
    For the former suggestion, we have increased the length of this section in the introduction. However, if a new chapter is created, it may affect the focus of our exposition, which is not quite in line with our chapter setup. For the latter, we have added a new section 5.1.
    Considering your former suggestion, we add this line in the introduction: " Ferrer-Font et al.[14] presented a spectral analyzer that enabled the concurrent analysis of up to 48 channels, thereby significantly enhancing the analytical capabilities of conventional flow cytometry systems.  "
    Considering your latter suggestion, we add this words in the section 5.1: "As controlled reaction chambers, high-throughput arrays, and positioning systems advance, microfluidics has amassed substantial data. Yet, not all data is readily analyzable. Thus, leveraging potent artificial intelligence for analysis becomes imperative, tightly fusing it with microfluidic devices to synergistically augment their capabilities. With the advent of machine learning, numerous advantages for particle counting based on microfluidic counters have emerged. Recent investigations have been focused on the development of machine and deep learning algorithms aimed at training models for the assessment and categorization of imaging flow cytometry images, thereby enhancing analytical workflows[146-148]. Machine learning offers the potential for optimizing parameters and models, significantly reducing processing time[149]. Moreover, contemporary research in the realm of flow cytometry encompasses cell classification, cell isolation, and their synergistic integration, benefiting from the integration of artificial intelligence techniques and microfluidic methodologies[150]. "
2. Another topic which may be included is the different types of lasers/white light excitation sources (commonly used wavelengths and its corresponding powers in terms of milliwatts) for fluorescence, photoacoustics, absorbance, etc. and the detectors which are used in different kinds of microfluidics-based counters. Since these equipments dictate a major proportion of the cost while setting up the overall instrument, such a topic will be very unique and valuable to the readers.
Answer: Thanks for your professional comments.
Just as you said, the excitation sources and detectors are the most important equipment for optical counters. However, we would like to focus on the efforts for improvement of sensitivity and throughput of counter. Therefore, we have not reviewed the equipment of counters.

3. A summary table should be provided which lists the most popular detection mechanisms, the smallest detectable concentrations, the most popular analytes which are detected, throughputs of detection, etc.
Answer: Thanks for your professional comments. 
    We want to acknowledge that creating such a table poses significant challenges due to the extensive and dynamic nature of the data involved. Nonetheless, to address your concern and provide valuable information to our readers, we have decided to present this information in the form of concise textual descriptions within the manuscript. This approach will allow us to provide a more nuanced and up-to-date perspective on these key aspects of microfluidic detection. 
    Considering your suggestion, we add Table 1. in the section 5: 
Table 1. Summary of the main types of microfluidic counters and their characteristics.
Classification    Name    Limit of detection    Integration Difficulty    Instrument price    Advantages    Disadvantages
Optical-Based Counters    Fluorescence Counter    30cfu∕ml (Escherichia coli O157:H7)    Medium    Moderately high    High sensitivity, multiple labeling    Complex equipment
    Optical Absorbed Counter    400nM(Rhodamine 6G concentration)    Low    Low    No labeling, easy to operate    Lower flux, low light transmission, poor effect
    Scattering Counter    1μm(Particle size)    High    High    Label-free, differentiates between different particles    Complex equipment, difficult to integrate
    Refractive Index Counter    4.3×〖10〗^7 mV/RIU(Refractive index change)    Low    Medium    Label-free, can monitor cell status    Mainly used for analytical testing, not suitable for opaque particles
Electrical-Based Counters    Resistance-Based Counter    1μm(Particle size)    Low    Low    Simple, label-free, low cost    Only suitable for conductive solutions, limited flux
    Capacitance-Based Counter    95μm(Bubble diameter)    High    Medium    Can be used in non-conductive solutions    Complicated electrode production, low sensitivity
    Inductance-Based Counter    11μm(Abrasive particle diameter)    Low    Low    Simple, low cost    Only detect metal abrasion particles, low throughput
    Impedance-Based Counter    6μm (Bollstein microsphere diameter)    Low    Low    Simple, low cost    Electric field may damage biological particles
Other Counters    Photoacoustic-Based Counter    No clear data    High    High    Label-free, real-time detection    Complex equipment, high cost
    Magnetic-Based Counter    20μm(Iron particle diameter)    Low    Low    Integrable, Low Cost    Only magnetic particles can be detected
    Thermal-Based Counter    90μm(Particle diameter)    Medium    Medium    Label-free, in-situ detection    Low sensitivity to small particles, susceptible to flow velocity and temperature

4. It would be nice to have a separate section on the current commercialization status of microfluidic devices. What are the applications of the present commercially available microfluidic devices, the major microfluidic service providers and the different challenges faced in this industry.
Answer: Thanks for your professional comments. 
    For your suggestion, we address the applications and challenges of currently available commercial microfluidic devices. in the new section 5:
5. Future Directions and Challenges
Significant progress has been made in developing various microfluidic systems. The main types of microfluidic counters and their characteristics are summarized by Table 1. Additionally, microfluidic technology's enhancement of experimental methods, cost reduction, and simplification has drawn broad attention in biotechnology. Microfluidic devices have diverse applications in life sciences, including real-time healthcare, precision and personalized medicine, regenerative medicine, prognosis, diagnostics, and treatment of tumor-related and non-tumor-related ailments. For instance, silver nanoparticles have gained notice due to their lack of microbial or viral resistance, making them useful for infection prevention and control[138]. Utilizing these traits, AV Blinov et al. have synthesized silver nanoparticles and oxidized variants, exploring their potential for suture coating components[139].
Micro-Electro-Mechanical Systems (MEMS) are intricate miniaturized devices often produced using microfabrication methods. They cleverly combine mechanical and electrical components to perform tasks akin to those accomplished by larger systems[140]. MEMS offer benefits like compact size, seamless integration, low weight, minimal power usage, and high resonant frequencies[141]. These attributes have led to growing interest in their use within the biomedical realm. Progress in computational technologies has facilitated the integration of microfluidic approaches into advanced MEMS device design, yielding advantages such as reduced energy consumption, limited reagent consumption, and enhanced detection sensitivity[142].
In biomedicine, MEMS technology has made notable strides and is recognized as BioMEMS. These devices are designed and manufactured for real-time disease diagnosis, biosensors, drug delivery systems, and surgical tools[143,144]. MEMS technology is widely utilized as a platform for producing enhanced and uniform nanoparticles. Additionally, wearable MEMS devices have become vital for individuals with chronic conditions, enabling remote monitoring of crucial signs like blood pressure, intracranial pressure, blood glucose levels, heart and respiratory rates, body temperature, and oxygen saturation[145].
Table 1. Summary of the main types of microfluidic counters and their characteristics.
Classification    Name    Limit of detection    Integration Difficulty    Instrument price    Advantages    Disadvantages
Optical-Based Counters    Fluorescence Counter    30cfu∕ml (Escherichia coli O157:H7)    Medium    Moderately high    High sensitivity, multiple labeling    Complex equipment
    Optical Absorbed Counter    400nM(Rhodamine 6G concentration)    Low    Low    No labeling, easy to operate    Lower flux, low light transmission, poor effect
    Scattering Counter    1μm(Particle size)    High    High    Label-free, differentiates between different particles    Complex equipment, difficult to integrate
    Refractive Index Counter    4.3×〖10〗^7 mV/RIU(Refractive index change)    Low    Medium    Label-free, can monitor cell status    Mainly used for analytical testing, not suitable for opaque particles
Electrical-Based Counters    Resistance-Based Counter    1μm(Particle size)    Low    Low    Simple, label-free, low cost    Only suitable for conductive solutions, limited flux
    Capacitance-Based Counter    95μm(Bubble diameter)    High    Medium    Can be used in non-conductive solutions    Complicated electrode production, low sensitivity
    Inductance-Based Counter    11μm(Abrasive particle diameter)    Low    Low    Simple, low cost    Only detect metal abrasion particles, low throughput
    Impedance-Based Counter    6μm (Bollstein microsphere diameter)    Low    Low    Simple, low cost    Electric field may damage biological particles
Other Counters    Photoacoustic-Based Counter    No clear data    High    High    Label-free, real-time detection    Complex equipment, high cost
    Magnetic-Based Counter    20μm(Iron particle diameter)    Low    Low    Integrable, Low Cost    Only magnetic particles can be detected
    Thermal-Based Counter    90μm(Particle diameter)    Medium    Medium    Label-free, in-situ detection    Low sensitivity to small particles, susceptible to flow velocity and temperature

Microfluidic devices face significant challenges in data analysis, emphasizing the integration of artificial intelligence for enhanced analytics. Material limitations, particularly with widely-used materials like PDMS, necessitate exploration of alternatives such as SEBS. Additionally, the manufacturing of microfluidic devices, especially MEMS devices, is beset by issues like fragility and contamination, requiring innovative solutions to improve their production processes. These multifaceted challenges collectively underscore the need for continuous advancement in microfluidic technology.
5.1 Data Analysis Capabilities
As controlled reaction chambers, high-throughput arrays, and positioning systems advance, microfluidics has amassed substantial data. Yet, not all data is readily analyzable. Thus, leveraging potent artificial intelligence for analysis becomes imperative, tightly fusing it with microfluidic devices to synergistically augment their capabilities. With the advent of machine learning, numerous advantages for particle counting based on microfluidic counters have emerged. Recent investigations have been focused on the development of machine and deep learning algorithms aimed at training models for the assessment and categorization of imaging flow cytometry images, thereby enhancing analytical workflows[146-148]. Machine learning offers the potential for optimizing parameters and models, significantly reducing processing time[149]. Moreover, contemporary research in the realm of flow cytometry encompasses cell classification, cell isolation, and their synergistic integration, benefiting from the integration of artificial intelligence techniques and microfluidic methodologies[150].
5.2 Material Defect
Polydimethylsiloxane (PDMS) is highly favored for microfluidic device manufacturing due to its rapid prototyping, user-friendliness, and biocompatibility advantages[151]. However, PDMS does have limitations. For instance, it's prone to absorbing small hydrophobic molecules, constraining manufacturing processes, parameters, and chip dimensions. Thus, the pursuit of alternative materials with superior performance is necessary. Recently, materials like styrene-butadiene-styrene (SEBS) gained wide application for their balanced mechanical properties, processability, and recyclability[152-154].
MEMS devices confront challenges in their overreliance on silicon and derivatives as exclusive materials[155]. This prompts materials science to boldly explore and develop to meet new material needs in microdevices.
5.3 Equipment Manufacturing
Numerous microfluidic devices encounter manufacturing challenges. MEMS devices, owing to their small size, fragility, and susceptibility to external influences, can experience problems like cracking, bending, or misalignment of moving parts[156]. Moreover, due to their complex mechanical structures, these devices are prone to issues arising from particle contamination, fatigue, fractures, friction, or surface adhesion[157].

5. In line 178-179, it is mentioned: “Additionally, the incorporation of optical components may be required, which can sometimes be expensive.” This needs clarification. What kinds of optical components are required for enhancing fluorescence signals and why is it expensive?
Answer: Thanks for your professional comments. 
    This sentence was intended to be a comparison between optical and electrical counters, but the placement at the end of the fluorescence technician may have been limiting, for which we deleted the sentence in its original position and preceded the first paragraph introducing the optical counter, i.e., the title of section 2.1.
    Considering your suggestion, we add this line after the title of section 2: " Optical-based counters are typically more costly compared to electrical-based counters. This is primarily attributed to the precision optical components required in optical systems, such as laser sources, filters, lenses, and high-performance detectors, including but not limited to Photomultiplier Tubes (PMTs) and Avalanche Photodiodes (APDs), which demand precise manufacturing and quality materials. Optical-based counters also involve intricate calibration and optimization to ensure accurate optical alignment and signal acquisition. Additionally, the need for fluorescent markers and high-resolution imaging equipment can further increase costs. Despite its higher cost, optical-based counters offer advantages such as high sensitivity, versatility, and non-destructive capabilities, making it suitable for various fields including biology, drug development, and environmental monitoring."
